



# Catchment Attributes and MEteorology for Large-Sample SPATially distributed analysis (CAMELS-SPAT): Streamflow observations, forcing data and geospatial data for hydrologic studies across North America

Wouter J. M. Knoben[1], Kasra Keshavarz[1], Laura Torres-Rojas[2], Cyril Thébault[1], Nathaniel W. Chaney[3], Alain Pietroniro[1], and Martyn P. Clark[1]

[1]Schulich School of Engineering, University of Calgary, Calgary, Alberta, Canada
[2]Atmospheric & Oceanic Sciences, Princeton University, Princeton, New Jersey, United States of America
[3]Civil and Environmental Engineering, Duke University, Durham, North Carolina, United States of America

**Correspondence:** Wouter Knoben (wouter.knoben@ucalgary.ca)

**Abstract.**

We present a new data set aimed at hydrologic studies across North America, with a particular focus on facilitating spatially distributed studies. The data set includes basin outlines, stream observations, meteorological data and geospatial data for 1426 basins in the United States and Canada. To facilitate a wide variety of studies, we provide the basin outlines at a lumped and

semi-distributed resolution; streamflow observations at daily and hourly time steps; variables suitable for running a wide range of models obtained and derived from different meteorological data sets at daily (1 data set) and hourly (3 data sets) time steps; and geospatial data and derived attributes from 11 different data sets that broadly cover climatic conditions, vegetation properties, land use, and subsurface characteristics. Forcing data are provided at their native gridded resolution, as well as averaged at the basin and sub-basin level. Geospatial data are provided as maps per basin, as well as summarized as catchment attributes

at the basin and sub-basin level with various statistics. Attributes are further complemented with statistics derived from the forcing data and streamflow, and have a particular focus on quantifying the variability of natural processes and catchment characteristics in space and time. Our goal with this data set is to build upon existing large-sample data sets and provide the means for more detailed investigation of hydrologic behavior across large geographical scales. In particular, we hope that this data sets provide others with the data needed to implement a wide range of modeling approaches, and to investigate the impact

of basin heterogeneity on hydrologic behaviour and similarity. The CAMELS-SPAT (Catchment Attributes and MEteorology for Large-Sample SPATially distributed analysis) is available at: https://dx.doi.org/10.20383/103.01216.

## 1  Introduction

Increases in geospatial data availability and computing power have enabled rapid advances in large-domain and large-sample hydrology (Cloke and Hannah, 2011; Addor et al., 2020). A key difference between these fields is the spatial continuity of the

study area. Where large-domain studies concern themselves with obtaining predictions across continuous areas, large-sample



studies tend to select separate basins within a given area of interest. The large-sample approach strikes a balance between spatial variability and ease of use. Large sample studies can be representative of larger spatial regions at a fraction of the computational effort needed to run a large-domain study over the same geographical region.

Building upon the foundations laid by the MOPEX data set (Schaake et al., 2006), a driving force behind the large-sample movement has been the "CAMELS" family of data sets. The original Catchment Attributes and MEteorology for Large-sample Studies (CAMELS) dataset was developed as a two-part initiative. First, basin-averaged meteorological time series were provided for several hundreds of basins across the Contiguous United States (Newman et al., 2015). Second, statistical descriptors (referred to as catchment attributes) of each catchment's hydroclimatic conditions were made available (Addor et al., 2017a). This combined data set has proven useful for various purposes, mainly within the overarching themes of understanding, quantifying and modeling hydrologic processes across a diverse range of catchments (e.g., Kratzert et al., 2019; Knoben et al., 2020; Stein et al., 2021) and quantifying hydrologic predictability (e.g., Wood et al., 2016; Newman et al., 2017). The success of the CAMELS dataset has motivated development of multiple (typically national) variants (see Table 1 for a summary of these), as well as the aggregated cloud-based CARAVAN collection (Kratzert et al., 2023, see also Färber et al. (2024)).

Table 1 provides a brief overview of the main characteristics of various CAMELS(-like) data sets. Because our interest is in hydrologic modeling, we limit this overview to data sets that include meteorologic time series that could serve as input to hydrologic models. A commonality between most of these data sets is a focus on aggregated data: meteorologic forcing data and catchment attributes are typically provided as basin-averaged values, and the temporal resolution of provided forcing data is almost always at daily time steps. Similarly, most datasets provide a specific selection of forcing variables: precipitation (P) and temperature (T) are always included, as well as a potential evapotranspiration (PET) time series or the variables necessary to calculate PET. In modeling terms, these data sets focus strongly on catchment modeling with lumped conceptual models. Such models treat catchments as single (i.e., lumped) entities, are typically run at daily time resolutions, and generally require only time series of P, T and PET to function. Commonly known examples of such models are SAC-SMA (National Weather Service, 2005), HBV (Lindström et al., 1997) and GR4J (Perrin et al., 2003). Such models are computationally cheap but often criticized for their somewhat empirical and spatially lumped nature, and their lack of explicit energy balance calculations.

Spatially-distributed process-based models, such as VIC (Hamman et al., 2018) and SUMMA (Clark et al., 2015a, b), address these concerns but come with the trade-off of increased computational cost and face their own challenges. Notable challenges include the definition of appropriate parameter values and questions about the scale-dependency of their constitutive functions (Hrachowitz and Clark, 2017). Investigating these models in large-sample studies could provide helpful insights, but running such models is not easily possible with most of the data sets listed in Table 1. The clearest exception to this are the LamaH-CE (Klingler et al., 2021) and LamaH-Ice data sets (Helgason and Nijssen, 2024), which cover the Upper Danube river basin in Central Europe and interior Iceland respectively. Both data sets provide data in a semi-distributed, spatially continuous fashion and provide a collection of forcing variables generally associated with process-based modeling approaches. However, the spatially continuous nature of these data sets means they are somewhat constrained geographically, covering an area of only 170,000 $\text{km}^2$ (roughly 600 by 300 km) in Central Europe and an area of 46,000 $\text{km}^2$ (roughly 300 by 150 km) in interior Iceland, respectively. Both datasets also still aggregate data at the sub-basin level, prohibiting the use of grid-based models.



There is a clear gap in the current collection of large-sample hydrologic data sets that (1) enables the use of spatially-distributed process-based models across a wide range of hydroclimatic conditions, and (2) enables studies aimed at investigating spatial heterogeneity at a resolution made possible by the geospatial data sets that underpin the current generation of large-sample hydrology data sets.

In this paper we introduce the CAMELS-SPAT data set ("Catchment Attributes and MEterology for Large-sample Studies for SPATially distributed analysis"). We expand on the original CAMELS data set (Newman et al., 2015; Addor et al., 2017a) in various ways. First, we provide data at native (i.e. gridded), sub-basin and basin levels, instead of treating each catchment only as a lumped entity. Second, we extend the geographical domain of the data set to include Canada, which includes various types of hydrologically challenging landscapes not included in the original CAMELS data set (e.g., glaciated basins, regions

with extensive permafrost, arctic deserts). Third, we provide a wider range of forcing variables at a temporal resolution (i.e., hourly) suitable for process-based modeling, in addition to a commonly used daily data set. Fourth, we provide a wider range of catchment attributes, with the specific goal of quantifying the attributes' ranges in time and space rather than providing mean values only. Compared to LamaH-CE and LamaH-Ice, our main contributions can be found in the wider range of hydroclimatic conditions found across the United States and Canada, and the inclusion of forcing and geospatial data at their

native (non-aggregated) resolution. Compared to HYSETS, another large-sample data set focused on North America, our main contributions can be found in the wider range of forcing variables, a higher temporal and spatial resolution of forcing data, and the inclusion of forcing and geospatial data at their native (non-aggregated) resolution.

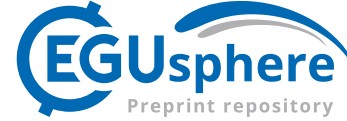



**Table 1.** Overview of large-sample data sets aimed at hydrologic modeling. Data sets are listed chronologically.

| Data set | Coverage Spatial Region | Coverage Temporal | # basins | Resolution Temporal | Resolution Spatial | Forcing data # products | Variables |
|---|---|---|---|---|---|---|---|
| MOPEX[1,2] | Contiguous US | 1948-2003 | 438 | Daily | Basin-averaged | Station observations within basins | Precipitation, climatic potential evaporation, maximum air temperature, minimum air temperature |
| CANOPEX[1] | Canada | Varies per basin | 698 | Daily | Basin-averaged | 2 (1 station, 1 gridded) | Precipitation, maximum temperature, minimum temperature |
| CAMELS[1] | Contiguous US | Varies per forcing dataset | 671 | Daily | Basin-averaged; per elevation band | 3 | Precipitation, maximum temperature, minimum temperature, shortwave downward radiation, day length, vapor pressure |
| CAMELS-CL[1] | Chile | Varies per basin | 516 | Daily | Basin-averaged | Multiple, depending on variable | Precipitation, maximum temperature, mean temperature, minimum temperature, potential evapotranspiration |
| HYSETS[1] | North America | Varies per basin | 14425 | Daily | Basin-averaged | 7 | Precipitation, maximum air temperature, minimum air temperature |
| CAMELS-BR[1] | Brazil | 1981-2018 | 897 | Daily | Basin-averaged | Multiple, depending on variable | Precipitation, maximum temperature, average temperature, minimum temperature, potential evapotranspiration, actual evapotranspiration |
| CAMELS-GB[1] | Great Britain | 1970-2015 | 671 | Daily | Basin-averaged | 1 | Precipitation, average temperature, potential evapotranspiration, potential evapotranspiration with interception correction, wind speed, specific humidity, downward shortwave radiation, longwave radiation |
| CABra[1] | Brazil | 1980-2010 | 785 | Daily | Basin-averaged | 3 | Precipitation, maximum temperature, minimum temperature, solar radiation, 2m wind speed, potential evapotranspiration (3 estimates), actual evapotranspiration |
| CAMELS-AUS[1] | Australia | Varies per forcing dataset and variable | 222 (v2: 561) | Daily | Basin-averaged | Multiple, depending on variable | Precipitation, maximum temperature, minimum temperature, potential evapotranspiration (4 estimates), actual evapotranspiration, solar radiation, vapor pressure, vapor pressure deficit, relative humidity at time of maximum temperature, relative humidity at time of minimum temperature, mean sea level pressure |
| LamaH-CE[1,3] | Central Europe | 1981-2019 | 859 | Daily, hourly | Basin-averaged at three basin levels | 1 | Precipitation, 2m air temperature, 10m wind in U-direction, 10m wind in V-direction, net solar radiation at the surface, net thermal radiation at the surface, surface pressure, total evapotranspiration |
| CCAM[1] | China | 1990-2020 | 4911 | Daily | Basin-averaged | 1 | Precipitation, 2m mean temperature, ground surface temperature, potential evapotranspiration, measured evaporation, ground pressure, relative humidity, 2m wind speed, sunshine duration |
| CAMELS-CH[1,4] | Switzerland and surrounding areas | 1981-2020 | 331 | Daily | Basin-averaged | 1 (used to derive extra variables) | Precipitation, maximum temperature, mean temperature, minimum temperature, relative sunshine duration |
| CAMELS-SE[1] | Sweden, with small parts of Norway | 1961-2020 | 50 | Daily | Basin-averaged | 1 | Precipitation, temperature |
| LamaH-ICE[1,3] | Iceland | Varies per basin | 107 | Daily, hourly | Basin-averaged at three basin levels | 3 | Precipitation, 2m air & dew point temperature, surface net solar & thermal radiation, surface pressure, specific humidity, complemented with various outputs from a land surface model such as soil water content, total evaporation and snow water equivalent |
| CAMELS-FR[1] | France | 1970-2021 | 654 | Daily | Basin-averaged | 1 | Solid precipitation, liquid precipitation, minimum & maximum air temperature, wind speed, specific humidity, atmospheric & visible radiation, 3 potential evapotranspiration estimates, as well as soil moisture and snow water equivalent estimates from a land surface model |
| CAMELS-DE[1] | Germany | Varies per basin | 1582 | Daily | Basin-averaged | 1 | Precipitation, minimum/mean/maximum temperature, humidity, radiation, potential evapotranspiration |
| CAMELS-DK[1] | Denmark | Varies per basin | 3330[5] | Daily | Basin-averaged | 1 | Precipitation, temperature, potential evapotranspiration, complemented with various outputs from a land surface model |
| CAMELS-IND[1] | India | 1980-2020 | 472[5] | Daily | Basin-averaged | Multiple, depending on variable | Precipitation, minimum/mean/maximum temperature, downward shortwave radiation, downward longwave radiation, wind speed in U, V and average direction, relative humidity, 2 potential evapotranspiration estimates, complemented with various outputs from a land surface model |

[1] References: MOPEX (Schaake et al., 2006), CANOPEX (Arsenault et al., 2016), CANOPEX (Arsenault et al., 2020), CAMELS (Newman et al., 2015; Addor et al., 2017a), CAMELS-CL (Alvarez-Garreton et al., 2018), HYSETS (Arsenault et al., 2020), CAMELS-BR (Chagas et al., 2020), CAMELS-GB (Coxon et al., 2020), CABra (Almagro et al., 2021), CAMELS-AUS (Fowler et al., 2021, 2024), LamaH-CE (Klingler et al., 2021), CCAM (Hao et al., 2021), CAMELS-CH (Höge et al., 2023), CAMELS-SE (Teutschbein, 2024), LamaH-ICE (Helgason and Nijssen, 2024), CAMELS-FR (Delaigue et al., 2024), CAMELS-DE (Loritz et al., 2024), CAMELS-DK (Liu et al., 2024), CAMELS-IND (Mangukiya et al., 2025).

[2] MOPEX forcing variables as currently available on https://hydrology.nws.noaa.gov/pub/gcip/mopex/US_Data/Us_438_Daily/.

[3] LamaH-CE and LamaH-ICE basins are spatially connected.

[4] CAMELS-CH forcing variables derived from the core forcing include: precipitation, mean temperature, global radiation, sunshine duration, wind speed, relative humidity, potential evapotranspiration, actual evapotranspiration, intercepted evapotranspiration

[5] CAMELS-DK provides streamflow observations for 304 out of 3330 basins; CAMELS-IND provides streamflow observations for 228 out of 472 basins.



## 2   Design considerations and outcomes

Our goal with this data set is to enable studies that investigate spatial heterogeneity across a wide variety of catchments, with
a specific focus on spatially-distributed process-based modeling. We also envision this data set to be used to compare the
performance of these models to their more empirical counterparts, and for analyses not directly based on hydrologic models.
Consequently, we processed a variety of data sources at various levels. We provide further detail about these requirements in
the following sub-sections, as needed. Our general methodology for creating CAMELS-SPAT is as follows:

1. Define an initial set of basins of potential interest, covering the United States and Canada;

2. Create consistent basin delineations for all basins identified under (1);

3. Obtain and process streamflow observations for the basins identified under (1), removing those basins for which no
   streamflow data can be found;

4. Obtain and process meteorological forcing data for the basins identified under (3);

5. Obtain and process geospatial data sets (e.g. data describing each basin's climate, vegetation, land use, topography, soil
   and geology) for the basins identified under (3);

6. Remove a number of very large basins from the basins identified under (3), and divide the remaining basins into various
   sub-datasets, based on disk space considerations;

7. Calculate catchment attributes using the data processed under (3), (4) and (5).

Figure 1 shows a visual summary of the main steps and decision points in this process, and each step is explained in more
detail in the following subsections. For the reader's benefit, we present combined descriptions of the methods and results for
each of these steps in the following seven subsections, instead of splitting these into dedicated Methods and Results sections.
The code used to generate this data set is available online (see "Code and Data Availability" statement, Section 6).

### 2.1   Basin preselection

#### 2.1.1   Context

We impose two initial constraints on the basins we will consider including in this data set. First, we have chosen to focus this
dataset on (near-)natural basins. Human impacts on the earth system are critically important but substantially complicate hy-
drologic behaviour and are typically difficult to quantify and thus difficult to account for during analyses. Such impacts include
but are not limited to: (i) the construction of water management structures such as dams and drainage ditches at the local level,
of which the location and size are difficult to ascertain and usually unreported in the continental scale data sets CAMELS-SPAT
relies on; (ii) the construction of large water management infrastructure such as diversions and reservoirs, which may appear in
continental scale data sets but for which operating procedures are typically unknown; (iii) surface and groundwater abstractions



**Figure 1.** Overview of the CAMELS-SPAT workflow. Grey boxes and light blue call-outs indicate specific folders on the GitHub repository, where the necessary code to reproduce these steps can be found. Note that repository folder *4_data_structure_prep* is not listed in this figure because it contains no methodological choices.

for e.g. agricultural and industrial use, for which abstraction and return volumes are typically unknown. That said, it is almost unavoidable that any selected basin includes at least some human impacts (tourism/recreation, drainage, forest management, etc.). We rely on existing classifications to select basins that are closer to the natural end of this continuum. Second, we require
the availability of at least some streamflow observations at a sub-daily resolution. Process-based models are typically run at sub-daily time steps to more accurately simulate diurnal variation in processes such as evaporation, transpiration, sublimation and snow melt. In certain basins such diurnal variability is visible in the streamflow record, and sub-daily observations are nec-





essary to evaluate the appropriateness of process-based model equations. Daily data is by definition too coarse to distinguish such patterns.

### 2.1.2 Methods and outcomes

For basins in the United States, we rely on the basin selection made by Newman et al. (2015) that was used for the CAMELS data set (Addor et al., 2017a). This ensures that some level of comparison between outcomes of studies using either CAMELS or CAMELS-SPAT is possible. We refer the reader to Section 2.1 in Newman et al. (2015) for a description of the criteria used to create this selection of 671 basins.

For basins in Canada, we start with the list of 1027 gauges included in the "Reference Hydrometric Basin Network" (RHBN, Environment and Climate Change Canada, 2020a, retrieved: 2022-08-18). These gauges have a minimum data availability of 20 years and minimal anthropogenic impacts as quantified by the presence of agriculture, built-up areas, and water management infrastructure, as well as population and road density. These criteria are comparable to those described in Newman et al. (2015). Note that agriculture presence in the Canadian prairie provinces (Alberta, Saskatchewan, Manitoba) and southern Ontario is substantial, and above the 10% area threshold used for the other provinces and territories (Pellerin and Nzokou Tanekou, 2020, p. 7). Excluding these basins would severely reduce the number Canadian gauges we could include in the data set, and we thus retain these gauges but include various data products in CAMELS-SPAT that can be used to quantify or filter by the presence of agriculture.

Our initial basin selection included 1698 basins across the United States and Canada. Various basins had to be removed due a lack of streamflow estimates or sub-daily data (see Section 2.3). We further removed several of the largest basins from the data set, under the assumption that any new insights that could be gained from these extremely large basins are minimal (especially given that these basins are severely under-gauged for their size) and do not outweigh the extra disk space needed to store the data for these basins (see Section 3 in the Supplementary Materials for details). Our final selection consists of 1426 basins, with an approximately even spread between the United States and Canada. For clarity, any outcomes shown in Sections 2.2 to 4.4 only show the final 1426 basins we have made publicly available, rather than the 1698 basins that are the outcome of this basin pre-selection step.

## 2.2 Basin delineation

### 2.2.1 Context

Hydrologic data sets such as this are conditional on having accurate basin outlines. Basin outlines are used to estimate a drainage basin's area, to crop meteorological and geospatial data to the area of interest, and to define the spatial extent of model configurations. Basin area estimates are also often used to convert the units of fluxes from volume-per-time to depth-per-time or vice versa (e.g. from $\mathrm{m^3\,s^{-1}}$ to $\mathrm{mm\,s^{-1}}$). Using incorrect basin area estimates can lead to large conversion errors that propagate into any further analysis (McMillan et al., 2023).





The basin polygons provided as part of the CAMELS data (Newman et al., 2014; Addor et al., 2017b) are administrative
boundaries. These polygons are not based on gauge locations, and the polygons thus tend to overestimate the basins' drainage
areas. Estimated area errors (derived from a comparison of reported upstream area for each gauge and actual area of the basin
polygon) are typically in the order of some percent (below 2% for approximately 70% of basins), but can be substantial (above
10% for approximately 8.5% of basins, with individual cases well above 100%). Additionally, openly available polygons for
the Canadian gauges did at the time of project initialization not fully cover all 1027 basins listed in the Reference Hydrometric
Basin Network (Environment and Climate Change Canada, 2020b, retrieved: 2022-01-31).

To address both concerns, we delineated new basin outlines for all basins identified as potential candidates in Section 2.1.
Our specific goals were to (1) identify the upstream area of each gauge, and (2) divide this upstream area into sub-basin
polygons of roughly equal size.

### 2.2.2 Method and outcomes

We obtained gauge metadata (location, name, reference areas, etc.), as well as reference basin outline polygons if these were
available, for all gauges identified in the first step. For the US gauges, metadata and polygons showing each basin's outline were
obtained from the CAMELS data set (Newman et al., 2014; Addor et al., 2017b). For the Canadian gauges, an initial download
of the Reference Hydrometric Basin Network (RHBN) metadata was used to identify which gauges are included in the RHBN
version released in 2020. Further metadata (location, name) were then extracted from the HYDAT database (Environment and
155 Climate Change Canada, 2010). Two different sets of reference polygons were available (Environment and Climate Change
Canada, 2020b; Government of Canada, 2022, accessed: 2022-08-23, 2022-08-18, respectively), of which we preferentially
used the newer polygons if these were available for our basins of interest.

To divide larger basins into smaller sub-basins we used the MERIT Basins data set (Lin et al., 2019). This data set contains
vectorized river basins and river networks, derived from the MERIT Hydro data (Yamazaki et al., 2019). The mean sub-basin
size in the MERIT Basins data is 45.6 $\mathrm{km}^2$ (median: 36.8 $\mathrm{km}^2$). We refer the reader to Lin et al. (2019) for further details.
We also obtained the MERIT Hydro flow direction and accumulation grids (Yamazaki et al., 2019). The MERIT Hydro data
is provided as gridded data in a regular longitude/latitude coordinate system (EPSG:4326). This is a common format (most
of the meteorological data and many of the geospatial data sets we discuss in Sections 2.4 and 2.5 are also only available
in EPSG:4326) and we adopt this as the standard in CAMELS-SPAT to the extent feasible. The one exception is raw RDRS
forcing data, which is natively provided on a custom rotated latitude/longitude grid. Any area calculations and certain shapefile
intersection operations are performed in the North America Albers Equal Area Conic projection (ESRI:102008) .

The MERIT Basin network was derived independently from gauges and the sub-basins in this data set therefore do not align
with gauge locations as reported by the United States Geological Survey and the Water Survey of Canada. For a given basin
we thus needed to clip the most downstream sub-basin polygon to the gauge. We therefore first mapped the gauge locations
onto the MERIT Hydro river network using automated techniques. This mapping is intended to guarantee that delineation of
the upstream are of a given gauge starts from a pixel in the flow direction grid that is part of the main river (rather than the most
downhill pixel of a single hillslope). However, there are various scenarios where automatic mapping is inaccurate and manual





intervention is needed. We identified those cases through a combination of accuracy metrics (area comparison between new basin delineation and reported reference area(s), and percentage overlap between new basin delineation and reference polygon if any were available), and visual inspection of the new basin delineation, reference polygon, underlying MERIT Hydro data grids, and satellite images. If necessary, we manually defined a better outlet location to delineate the basin from and tracked this intervention in the CAMELS-SPAT metadata. We also assigned confidence ratings to our new basin polygons based on these quality assurance checks. As the final step, we identified all cases of nested gauges where a larger basin includes a smaller one. In such cases we split the sub-basin polygon that contains the nested gauge and assign unique identifiers to the upstream and downstream parts of the sub-basin and river segment.

Figure 2 shows the resulting polygons for the 1426 basins that form the final CAMELS-SPAT data set, with colors indicating our confidence ratings. "Unknown" refers to cases where no confidence rating could be assigned, mainly due to lacking reference polygons. "Low" ratings are assigned when evidence suggests that our basin delineations are inaccurate and we were unable to manually find a better outlet location that would lead to improved basin outlines. "Medium" ratings indicate that there are substantial differences between our new delineations and existing ones and/or reference areas, but that it is difficult to decide whether our new delineation or the reference(s) are more accurate. "High" ratings are assigned when there is a clear match between our new polygons and the reference(s), or when evidence suggests our new delineations are more accurate than the reference(s). Detailed reasons for these ratings are tracked as part of the CAMELS-SPAT metadata. Medium and low confidence ratings occur primarily in regions with flat topography where finding the true outline of any drainage basin is difficult.

## 2.3 Streamflow observations

### 2.3.1 Context

Streamflow is a key variable for many hydrologic studies. Streamflow estimates are typically provided as either instantaneous values (i.e., valid at a given point in time) or as averages over a given time interval. It is critical to know what type of values (instantaneous or time-averages) are available, as well as the time zones data are provided in.

The United States Geological Survey (USGS) typically collects instantaneous streamflow observations at 15- or 60-minute intervals. USGS also provides daily average values, computed from the instantaneous data from 00:00 to 24:00 Local Standard Time (LST; USGS, personal communication, 2023-06-20). Both instantaneous values and daily averages are publicly available.

The Water Survey of Canada (WSC) typically collects instantaneous streamflow observations at 5-minute intervals, and from these calculates daily averages that are reported in LST through the HYDAT database (WSC, personal communication, 2023-07-04). However, when instantaneous values are extracted through the WSC API, the time series are converted to UTC before being given to the user (Government of Canada, accessed: 2023-12-22). Instantaneous streamflow observations are publicly available for the period between present and minus 18 months. Daily average values are available for the full time period for which a gauge has been active.

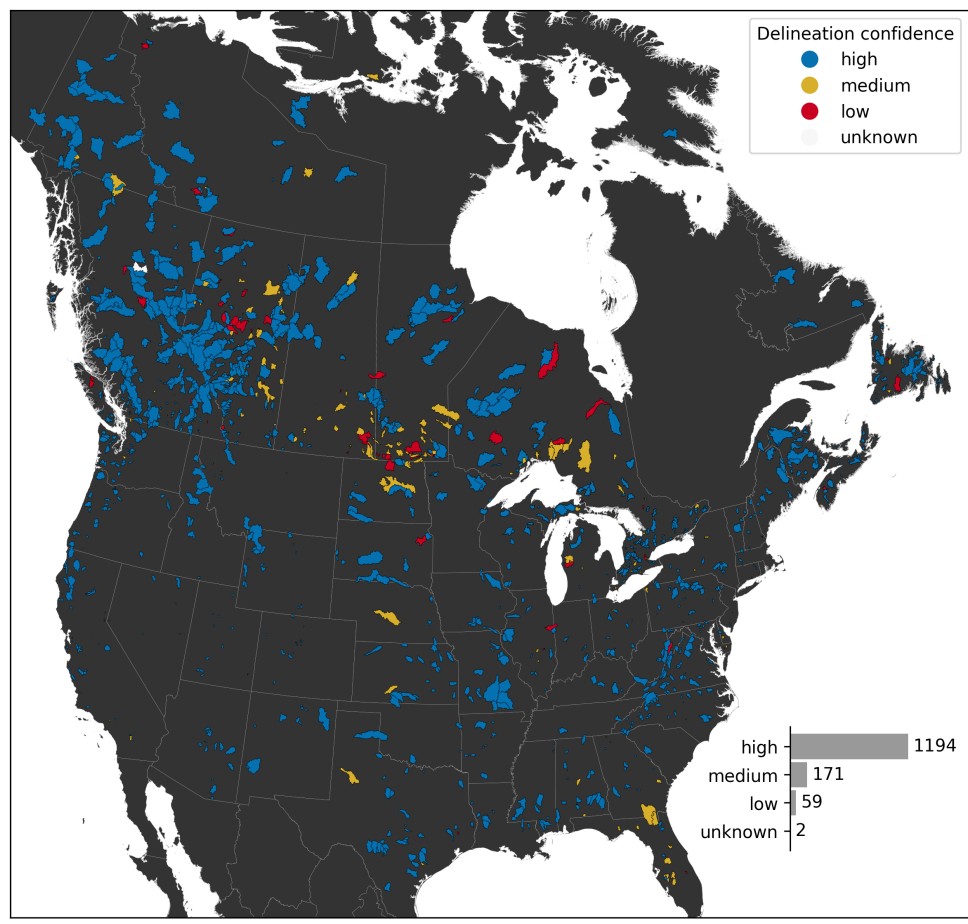

**Figure 2.** Location and delineation confidence of 1426 CAMELS-SPAT basins. Political boundaries by Commission for Environmental Cooperation (2022, accessed 2023-12-20)

.

Our goal with this project is to provide data useful for running and evaluating process-based hydrological models. We therefore include daily average streamflow values as available through USGS and WSC. We also include hourly average streamflow values to match the temporal resolution of our selected meteorological data sets. Hourly average flow data are computed from the sub-daily instantaneous data available through both agencies. All flow data, as well as meteorological forcing data, are included in the CAMELS-SPAT data set in Local Standard Time. The timezone of each gauge is tracked as

part of the meta data.



### 2.3.2 Method and outcomes

For the gauges in the United States, daily average streamflow data and instantaneous (sub-daily) data can both be extracted through API requests (https://nwis.waterservices.usgs.gov/nwis/dv/ and https://nwis.waterservices.usgs.gov/nwis/iv/, respectively; accessed 2023-06-16). For the Canadian gauges, sub-daily data were extracted from the Environment and Climate

Change Canada Web Service Links Interface (https://wateroffice.ec.gc.ca/services/links_e.html; accessed 2023-04-05). Daily data were extracted from the HYDAT database, version *20230505*. We excluded 4 gauges in the United States, as well as 180 Canadian gauges from the original 1697 preselected stations because sub-daily data was not available for these stations. We removed a further 13 Canadian gauges for lacking daily discharge values. Manual checks of these gauges through the WSC website (https://wateroffice.ec.gc.ca/search/historical_e.html; last access: 2025-02-06) indicate that these stations are

measuring water levels in lakes.

Daily average values for both countries are provided in Local Standard Time (LST). We updated the time indices for the sub-daily instantaneous values to match. For the gauges in the USA, this meant shifting the time series by 1 hour for time steps that were provided in local daylight saving time for gauges in states where daylight saving time is observed. For the Canadian gauges, this meant shifting the entire time series for each gauge by the offset needed to convert UTC to LST. We then set

any negative streamflow values to zero, and used a mass-conserving averaging approach to turn instantaneous flow data into hourly averages (see Section 1 in the Supplementary Materials for more details about the averaging procedure). We specified the condition that every hourly average must be based on at least one observation during that time window. Hours for which no data observations were available were set to Not-a-Number (NaN).

Note the critical assumption that we calculated the average hourly flows as the value at the full hour (e.g., 12:00) using a

230 forward-looking window (i.e., in this case the value at 12:00 is the average during the time window 12:00-13:00). This matches the daily flows, which are provided under the same assumption by USGS and WSC (e.g., the Jan-1 2000 value is calculated from data between 00:00 Jan-1 and 24:00 Jan-1; USGS, personal communication, 2023-06-20; WSC, personal communication, 2023-06-26). This information is also stored in the *time_bnds* (time bounds) variable available in the provided NetCDF files.

Daily and sub-daily observations were originally provided in text-based formats. We converted these to NetCDF4 formats,

to ensure consistency between gauges in the two countries and to track metadata in a more accessible way (compared to storing the metadata in separate files or headers in text files). For both USGS and WSC data we retained the quality flags that accompany the data and stored these in the same NetCDF files that contain the streamflow observations. These quality flags indicate conditions that may adversely affect the observations (e.g., gauge malfunction, ice conditions) and whether data has been formally approved or is still considered provisional.

Figure 3 shows aggregated flow data availability for the 1426 catchments included in the CAMELS-SPAT data set. Hourly flow data comes in two distinct categories: short (< 2 years) records for the Canadian gauges and much longer records for gauges located in the United States. This is a consequence of Water Survey of Canada's policy to make high-resolution gauge data only publicly available for a short historical period. Missing data for these shorter records are however typically low (see also Fig. A1). For approximately 80% of gauges, missing hourly observations account for up to 10% of record length. Data




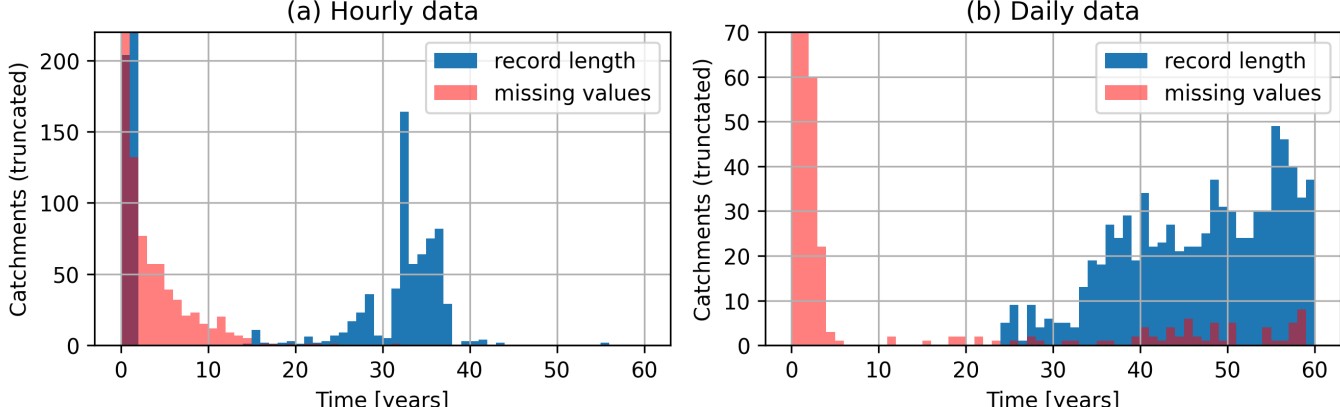

**Figure 3.** Flow data availability for gauges included in CAMELS-SPAT. Record length refers to the period between the first publicly available flow record for a given station, and its last. Missing values occur within this record period and are given here in the same units as the record length itself. Note that both y-axes are truncated: in (a), *missing values* has a count of 913 for *time* is [0,1], and *record length* has a count of 560 for *time* = [1,2]; in (b), *missing values* has counts of 1156 and 112 for *time* is [0,1] and [1,2], respectively.

may be missing for up to 40% of the record for most remaining gauges, with a handful of gauges having extremely large data gaps. Daily data record lengths are similar for Canadian and United States gauges. Missing values are relatively rare (<10% for up to 1350 out of our 1426 gauges), though can be substantial (up to 80 to 95% for the remaining gauges; see Fig. A1). The period with the greatest overlap of data records is 1990-2020; hourly observations are available for only a handful of gauges before this time.

## 2.4 Forcing data

### 2.4.1 Context

Meteorological forcing data in existing data sets is typically provided as catchment-averaged (lumped) daily data, and tends to be limited to precipitation, temperature and potential evapotranspiration variables (Table 1). While a large number of the more conceptual models can be run with only precipitation, temperature and potential evapotranspiration inputs (see e.g., Knoben et al., 2019; Trotter et al., 2022), more complex hydrologic models typically require a wider array of inputs at higher temporal resolution. Table 2 shows a brief overview of meteorological data requirements for a selection of process-based hydrological models. Typical variables include (1) precipitation, (2) air temperature, (3) radiation terms, often distinguishing between shortwave and longwave radiation, (4) air pressure, (5) humidity, and (6) wind speed.

It is clear from Table 2 that it is impossible to define a small set of forcing variables that would allow the use of a large number of process-based hydrologic models. We therefore decided to include a broad selection of meteorological variables, accepting that this comes at the cost of extra disk space. We provide these variables at hourly time steps, at their native gridded resolution as well as averaged at the sub-basin level. To facilitate the use of the broadest range of modeling tools we also





include time series of potential evaporation (see footnote in Table 3) and forcing variables aggregated at the lumped basin level.



**Table 2.** Meteorological data needs for CATFLOW (Maurer and Zehe, 2007), CHM (Marsh et al., 2020), CHRM (Pomeroy et al., 2007), ES-CROC (Lafaysse et al., 2017), HYPE (SMHI, 2022), MESH (Mekonnen and Brauner, 2020), Noah LSM (Mitchell et al., 2005), PARFLOW (Maxwell et al., 2019), MM-PIHM (PIHM team, 2007; Yuning Shi, 2018), SUMMA (Clark et al., 2015a, b; Nijssen, 2017), VIC (Liang et al., 1994; Hamman et al., 2018) and WaSIM (Schulla, 2021). Models are listed alphabetically. Optional inputs indicated with *. t indicates an arbitrary time unit.

| Variable | CATFLOW | CHM | CHRM | ES-CROC | HYPE | MESH |
|---|---|---|---|---|---|---|
| Precipitation | $[\mathrm{m\,t^{-1}}]$ | $[\mathrm{mm\,t^{-1}}]$ | $[\mathrm{mm\,t^{-1}}]$ | $[\mathrm{kg\,m^{-2}\,s^{-1}}]$ | $[\mathrm{mm\,t^{-1}}]$ | $[\mathrm{kg\,m^{-2}\,s^{-1}}]$ |
| Downward shortwave radiation | $[\mathrm{W\,m^{-2}}]$ | $[\mathrm{W\,m^{-2}}]$ | | $[\mathrm{W\,m^{-2}}]$ | $[\mathrm{MJ\,m^{-2}\,d^{-1}}]$* | $[\mathrm{W\,m^{-2}}]$ |
| Downward longwave radiation | | $[\mathrm{W\,m^{-2}}]$ | | $[\mathrm{W\,m^{-2}}]$ | | $[\mathrm{W\,m^{-2}}]$ |
| Air temperature | [C] | [C] | [C] | [K] | [C] | [K] |
| Air pressure | | | | [Pa] | | [Pa] |
| Specific humidity | | | | | | $[\mathrm{kg\,kg^{-1}}]$ |
| Wind speed (U-direction) | | | | | $[\mathrm{m\,s^{-1}}]$* | |
| Wind speed (V-direction) | | | | | $[\mathrm{m\,s^{-1}}]$* | |
| Sunshine duration | | | | | | |
| Reflected shortwave radiation | | | $[\mathrm{W\,m^{-2}}]$ | | | |
| Net radiation | $[\mathrm{W\,m^{-2}}]$ | | $[\mathrm{W\,m^{-2}}]$ | | | |
| Vapor pressure | | | | | | |
| Relative humidity | [%] | [%] | [%] | [%] | $[-]$* | |
| Wind speed (mean) | $[\mathrm{m\,s^{-1}}]$ | $[\mathrm{m\,s^{-1}}]$ | $[\mathrm{m\,s^{-1}}]$ | $[\mathrm{m\,s^{-1}}]$ | $[\mathrm{m\,s^{-1}}]$ | $[\mathrm{m\,s^{-1}}]$ |
| Wind direction | [degrees] | [degrees] | | | | |
| **Variable** | **Noah LSM** | **PARFLOW** | **MM-PIHM** | **SUMMA** | **VIC** | **WaSIM** |
| Precipitation | $[\mathrm{inch\,30min^{-1}}]$ | $[\mathrm{mm\,s^{-1}}]$ | $[\mathrm{kg\,m^{-2}\,s^{-1}}]$ | $[\mathrm{kg\,m^{-2}\,s^{-1}}]$ | $[\mathrm{mm\,t^{-1}}]$ | [mm] |
| Downward shortwave radiation | $[\mathrm{W\,m^{-2}}]$ | $[\mathrm{W\,m^{-2}}]$ | $[\mathrm{W\,m^{-2}}]$ | $[\mathrm{W\,m^{-2}}]$ | $[\mathrm{W\,m^{-2}}]$ | $[\mathrm{Wh\,m^{-2}}]$ |
| Downward longwave radiation | $[\mathrm{W\,m^{-2}}]$ | $[\mathrm{W\,m^{-2}}]$ | $[\mathrm{W\,m^{-2}}]$ | $[\mathrm{W\,m^{-2}}]$ | $[\mathrm{W\,m^{-2}}]$ | |
| Air temperature | [C] | [K] | [K] | [K] | [C] | [C] |
| Air pressure | [mbar] | [Pa] | [Pa] | [Pa] | [kPa] | |
| Specific humidity | | $[\mathrm{kg\,kg^{-1}}]$ | | $[\mathrm{g\,g^{-1}}]$ | | |
| Wind speed (U-direction) | | $[\mathrm{m\,s^{-1}}]$ | | | | |
| Wind speed (V-direction) | | $[\mathrm{m\,s^{-1}}]$ | | | | |
| Sunshine duration | | | | | | $[-]$ |
| Reflected shortwave radiation | | | | | | |
| Net radiation | | | | | | |
| Vapor pressure | | | | | [kPa] | |
| Relative humidity | $[-]$ | | [%] | | | $[-]$ |
| Wind speed (mean) | $[\mathrm{m\,s^{-1}}]$ | | $[\mathrm{m\,s^{-1}}]$ | $[\mathrm{m\,s^{-1}}]$ | $[\mathrm{m\,s^{-1}}]$ | $[\mathrm{m\,s^{-1}}]$ |
| Wind direction | | | | | | |





 **2.4.2   Methods and outcomes**

CAMELS-SPAT includes four forcing data sets, each with a specific focus:

1. First, we primarily use the high-resolution RDRS v2.1 data set (Gasset et al., 2021, available at 10km or approximately 0.09° resolution). RDRS covers the North American continent and provides those variables needed to run process-based models directly and derive most other variables listed in Table 2. A key advantage of RDRS is that it assimilates
precipitation observations.

2. Second, for continuity with the original CAMELS data set, we include the Daymet v4 R1 data set (Thornton et al., 2021, available at 1km or approximately 0.009° resolution). Daymet is based on weather station observations and gridded terrain data, and available at daily resolution between 1980 and 2023 on a 365-day calendar (i.e., during leap years December $31^{st}$ is missing). The data set does not include all the forcing variables needed to run process-based models, 275   but, if combined with an appropriate estimate of potential evapotranspiration (PET), provides sufficient information to run more conceptual and data-driven models. We infill the missing day in leap years as a linearly interpolated value between the preceding and following day. Following Newman et al. (2015), we add a Priestley-Taylor PET estimate (Priestley and Taylor, 1972, further details available in the Supplementary Materials).

3. Third, to facilitate possible extension of CAMELS-SPAT beyond North America, as well as provide hourly data for 280   gauges with observations before 1980 (i.e., outside the time period covered by RDRS), we include the globally available ERA5 data (Hersbach et al., 2020, available at 0.25° resolution). Like RDRS, ERA5 provides all variables needed to run process-based models directly, and derive most other variables listed in Table 2. However, unlike the other data sets listed here, ERA5 is a reanalysis product and does not integrate station observations. Local accuracy may thus be lower for ERA5 data than for data sets that do use station observations.

4. Fourth, to partly address this weakness of ERA5 data, we include the high-resolution EM-Earth data set (Tang et al., 2022b, available at 0.10° resolution). Previous work has shown that using station-based precipitation and temperature data from EM-Earth provides better modeling results for our area of interest than using ERA5 alone (Rakovec et al., 2023). However, note that the EM-Earth has a fixed temporal coverage of 1950-2019, whereas our selected gauges have data beyond 2019.

Table 3 shows an overview of forcing variables available as time series in the CAMELS-SPAT data set. Compared to Table 2, we provide net radiation terms at the surface separated into net shortwave and net longwave terms, and do not provide a summed net radiation component nor a reflected shortwave variable. Either can be easily derived from the provided net shortwave and longwave components (see Hogan (2015), but also footnote 2 in Table 3). We also do not provide sunshine duration because this is not available in RDRS, Daymet and EM-Earth. While sunshine duration is available in ERA5, it is not 295   an independent variable: it is derived directly from downward shortwave radiation using a threshold of $120 \, \mathrm{W \, m^{-2}}$ (Hogan, 2015). We complement the forcing data sets with various additional variables derived from the downloaded data in cases where





we judged the processing to be too cumbersome to pass down to the user (i.e., vapor pressure, relative humidity, wind direction), or the variable seemed to be of general interest (i.e., mean wind speed, PET). Potential evapotranspiration estimates for Daymet were derived using the Priestly-Taylor formula (Priestley and Taylor, 1972); PET estimates for RDRS were derived using the

300 FOA-56 Penman-Monteith method (Allen et al., 1998). The equations used to derive data are provided in the Supplementary Materials. While this list of variables is unlikely to completely cover all models' data needs, it will provide a reasonable starting point for a large number of models.

We retained the original variable names used in each data set so that users may easily refer to the existing documentation of RDRS, Daymet, ERA5 and EM-Earth if needed. For convenience and simplicity from a user perspective, we converted all data

to use a consistent set of units. This is mostly straightforward but required an assumption for the density of water which we set at $1000 \, \mathrm{kg \, m^{-3}}$. Data are provided for the full time period covered by the observational record of each individual gauge when possible, including time steps for which streamflow data are missing (see also Section 4.2.1 and Table 4). For all variables, metadata (descriptions, units, derivations if applicable) are stored as variable attributes in the NetCDF files.

We provide the forcing data at three different spatial aggregation levels: (1) as gridded values at the native resolution of each

310 data set, clipped to the basin outline; (2) aggregated at the sub-basin level; (3) aggregated at the basin level (i.e., the level at which most of the data sets listed in Table 1 provide data). Averaging of the gridded data to (sub-)basin polygons was done with the EASYMORE toolbox (Gharari et al., 2023a).

RDRS, ERA5 and EM-Earth provide data at hourly resolution, in Coordinated Universal Time (UTC). We process these time indices to be in each gauge's Local Standard Time (LST) instead, so that the time indices in the forcing file align with those used

for the flow observations. We make a slight adjustment for the 57 basins that are located in regions following Newfoundland Standard Time (NST [UTC − 3h30], National Research Council Canada (2019)). All forcing data products are only available at whole hours, and thus cannot easily be converted to NST. We treat these basins as following Atlantic Standard Time (AST [UTC − 4h00]) instead. Note that this leads to a 30-minute offset between forcing data and streamflow observations for these basins. Daymet data is already provided as daily average values calculated in LST and requires no further adjustment.

Variables in these forcing data sets are either instantaneous (i.e., representative of conditions at a specific point in time) or time-averaged (i.e., representative of conditions over a given time window), and this means the time stamps in each NetCDF file must be interpreted differently for different variables. For any instantaneous variable, a value is valid at the specific moment in time given by the time stamp (European Centre for Medium-range Weather Forecasting, 2023c). For any time-averaged variables, we need to distinguish between two cases. RDRS and ERA5 use *period-ending* or *backward-looking* time stamps,

meaning that, for example, the average precipitation rate at time 12:00 is the average rate over the interval 11:00-12:00 (N. Gasset, personal communication, 2024; European Centre for Medium-range Weather Forecasting, 2023b, Section: "Mean rates/fluxes and accumulations"). EM-Earth's precipitation variable instead uses *period-beginning* or *forward-looking* time stamps, meaning that, for example, the average precipitation rate at time 12:00 is the average rate over the interval 12:00-13:00 (G. Tang, personal communication, 2024). Table 3 provides an overview of all forcing variables and summarizes this

information.



**Table 3.** CAMELS-SPAT meteorological variables. Variable names shown in bold indicate derived variables. "Flux validity" indicates how time-averaged variables must be interpreted.

| | Data set | RDRS | Daymet | ERA5 | EM-Earth |
|---|---|---|---|---|---|
| | Resolution | Hourly | Daily | Hourly | Hourly |
| | Flux validity | Period-ending[1] | n/a | Period-ending[2] | Period-beginning[3] |
| **Time-averaged variables** | Units | Name in NetCDF files | | | |
| Precipitation rate | $[\mathrm{kg\,m^{-2}\,s^{-1}}]$ | RDRS_v2.1_A_PR0_SFC | prcp | mtpr | prcp |
| Downward shortwave radiation | $[\mathrm{W\,m^{-2}}]$ | | srad | msdwswrf | |
| Downward longwave radiation | $[\mathrm{W\,m^{-2}}]$ | | | msdwlwrf | |
| Net surface shortwave radiation | $[\mathrm{W\,m^{-2}}]$ | | | msnswrf[4] | |
| Net surface longwave radiation | $[\mathrm{W\,m^{-2}}]$ | | | msnlwrf[4] | |
| Potential evapotranspiration rate | $[\mathrm{kg\,m^{-2}\,s^{-1}}]$ | | **pet** | mper[5] | |
| **Instantaneous variables** | Units | Name in NetCDF files | | | |
| Downward shortwave radiation | $[\mathrm{W\,m^{-2}}]$ | RDRS_v2.1_P_FB_SFC | | | |
| Downward longwave radiation | $[\mathrm{W\,m^{-2}}]$ | RDRS_v2.1_P_FI_SFC | | | |
| Potential evapotranspiration rate | $[\mathrm{kg\,m^{-2}\,s^{-1}}]$ | **pet** | | | |
| Air temperature | $[\mathrm{K}]$ | RDRS_v2.1_P_TT_1.5m | | t | tmean |
| Minimum daily air temperature | $[\mathrm{K}]$ | | tmin | | |
| Maximum daily air temperature | $[\mathrm{K}]$ | | tmax | | |
| Daylight length | $[\mathrm{s\,day^{-1}}]$ | | dayl | | |
| Air pressure | $[\mathrm{Pa}]$ | RDRS_v2.1_P_P0_SFC | | sp | |
| Specific humidity | $[\mathrm{kg\,kg^{-1}}]$ | RDRS_v2.1_P_HU_1.5m | | q | |
| Relative humidity | $[\mathrm{kPa\,kPa^{-1}}]$ | RDRS_v2.1_P_HR_1.5m | | **rh** | |
| Vapor pressure | $[\mathrm{kPa}]$ | **e** | vp | **e** | |
| Wind speed (U-direction) | $[\mathrm{m\,s^{-1}}]$ | RDRS_v2.1_P_UUC_10m | | u | |
| Wind speed (V-direction) | $[\mathrm{m\,s^{-1}}]$ | RDRS_v2.1_P_VVC_10m | | v | |
| Wind speed (mean) | $[\mathrm{m\,s^{-1}}]$ | RDRS_v2.1_P_UVC_10m | | **w** | |
| Wind direction | $[\mathrm{degrees}]$ | **phi**[6] | | **phi**[6] | |

[1] N. Gasset, personal communication, 2024.

[2] See: https://confluence.ecmwf.int/pages/viewpage.action?pageId=82870405#ERA5:datadocumentation-Table4 (last access: 2024-01-03), https://confluence.ecmwf.int/pages/viewpage.action?pageId=82870405#ERA5:datadocumentation-Table9 (last access: 2024-01-03), https://confluence.ecmwf.int/pages/viewpage.action?pageId=82870405#ERA5:datadocumentation-Table2 (last access: 2024-01-03).

[3] G. Tang, personal communication, 2024.

[4] Note that these net radiation terms are based on interactions between the atmospheric and land surface components of the ERA5 modeling chain, and should thus only be used carefully as model input to prevent cases where the user's model duplicates processes already accounted for by the ERA5 models.

[5] Assumptions underlying this variable are described here: https://codes.ecmwf.int/grib/param-db/?id=228251 (last access: 2024-01-01). Note that we provide the equivalent variable as a mean rate as part of the CAMELS-SPAT data, but the URL for that variable lacks a clear description: https://codes.ecmwf.int/grib/param-db/?id=235070 (last access: 2024-01-01).

[6] We derived most additional variables before averaging the gridded data onto (sub-)basins, but this is not easily possible for wind direction. Instead, we calculate wind direction separately for the gridded, semi-distributed and lumped cases from u- and v-components after (sub-)basin averages of these variables were created. We use the meteorological wind direction as defined by ECMWF (European Centre for Medium-range Weather Forecasting, 2023a): wind direction in this case indicates the direction the wind comes from, not where it goes.





## 2.5 Geospatial data

### 2.5.1 Context

Geospatial data in existing data set covers four broad categories: (1) meteorology (as time series and derived summary statistics), (2) vegetation and land use; (3) topography; (4) soil and geology. In the current large-sample data sets, geospatial data
are typically not provided as maps in their original formats but tend to be presented as spatial statistics (mean, mode, etc.). These statistical summaries of the original data, commonly referred to as catchment attributes, can be helpful to succinctly characterize a location's hydroclimatic conditions and support classification efforts. For modeling purposes, geospatial data play a key role in defining model configurations and parameter values. For example, models such as Noah-LSM (Niu et al., 2011) and SUMMA (Clark et al., 2015a, b) rely on vegetation and soil classes to provide initial values for a number of land
use and soil parameters. More generally, models might require the height of the vegetation canopy in the vertical direction, or the fraction of the basin covered by open water in the horizontal direction as inputs. It is practically impossible to cover all possible use case, and we therefore provide the geospatial data as maps clipped to the basin outlines. The maps will allow users to derive model parameters and further catchment delineations (such as elevation zones, or land cover polygons), and to derive additional catchment attributes if our existing selection of attributes does not cover a particular study's needs (see Section 3).
Figure 4 shows an overview of the 11 different data sets we selected for use in CAMELS-SPAT.

### 2.5.2 Methods and outcomes

For internal consistency of the CAMELS-SPAT data, we selected various geospatial data sets that cover at least the United States and Canada. The specific processing steps vary, but in general processing for each data set involved downloading the data at continental or larger scales and clipping the data to the basin polygons. We also ensured all geospatial maps are provided
in a regular latitude/longitude coordinate system (EPSG:4326). Figure 4 provides an overview of the geospatial data layers, using a single basin as an example.

**Climate:** Long-term monthly means of several climate variables can be obtained from the WorldClim data set (Fick and Hijmans, 2017). The advantage over calculating these means from gridded forcing data is WorldClim's much higher spatial resolution. Available variables are long-term means computed from 30 years each, showing minimum, mean and maximum
monthly temperature, as well as monthly precipitation, solar radiation, wind speed and water vapor pressure. WorldClim's data license does not allow redistribution of their raw data, but does allow the data to be used to calculate derived statistics and redistribute those. We primarily use the WorldClim data to calculate various attributes that quantify the spatial heterogeneity in climatic conditions, and include various derived maps as part of CAMELS-SPAT.

**Vegetation:** Process-based hydrological models typically include explicit representations of vegetation cover in a catchment.
CAMELS-SPAT includes two data sets from which vegetation parameters may be derived. First, we included time series of Leaf Area Index (LAI) observations, derived from MODIS satellite observations (Myneni et al., 2021, MCD15A2H.061). These observations are available at an 8-day temporal resolution and cover the period 2002-07-04 to 2023-10-08. Certain models may be able to ingest these maps directly, or typical seasonal LAI patterns may be derived from them. In addition,



we included estimates of forest height in 2000 and 2020 (Potapov et al., 2021, part of the Global Land Cover and Land Use
Change, 2000-2020 data).

**Land cover and land use:** To further assist parametrization and classification efforts, we included three different products
related to land cover and land use. First, the Landsat-Derived Global Rainfed and Irrigated-Cropland Product (LGRIP30,
Thenkabail et al., 2021; Teluguntla et al., 2023) can be used to estimate the magnitude and type of agriculture practiced in each
basin. Second, we include a map of International Geosphere–Biosphere Programme (IGBP) land classes in each basin, derived
from MODIS satellite observations (Friedl and Sulla-Menashe, 2022). Third, we include high-resolution Global Land Cover
and Land Use 2019 maps (Hansen et al., 2022). This is very high-resolution data derived from Landsat satellite observations,
used to classify the landscape into several broad categories (inland water, permanent snow and ice, cropland, built-up, terra
firma and wetlands) with several of these consisting of subclasses based on build-up area extent, and vegetation extent and
height.

**Open water:** We include cutouts of the HydroLAKES data (Messager et al., 2016) to quantify the extent, type and volumes
of open water bodies in each basin. This data can be used to estimate each catchment's open water area, retention volumes and
parametrization of reservoir and lakes modules in hydrologic and/or routing models.

**Topography:** The MERIT Hydro Digital Elevation Model (DEM) used for basin delineation (Yamazaki et al., 2019) is
also part of the maps provided for each catchment. We used the DEM to derive separate maps of slope and aspect because of
380 their hydrologic relevance. For both, the DEM was first reprojected into ESRI:102009 (NAD 1983 Lambert North America)
to ensure consistency between horizontal and vertical units. We then calculated slope maps expressed as angles (i.e., degrees),
and aspect maps in degrees indicating which direction a slope faces (with 0/90/180/270º being North/East/South/West-facing
slopes respectively). Additional variables such as elevation bands may be derived from the DEM map, but due the subjectivity
involved in deciding where the boundaries between the elevations bands are we have not done so. The DEM data can may also
be useful to apply elevation-dependent lapse rates to meteorologic variables.

**Soil and geology:** We provide maps from three different data sets to characterize each catchment's subsurface. First, SOIL-
GRIDS 2.0 (Poggio et al., 2021) provides estimates of various soil properties (bulk density, percentage coarse fragments,
organic carbon content, and sand, silt and clay percentages) at six different depths (0-5 cm, 5-15 cm, 15-30 cm, 30-60 cm, 60-
100 cm, 100-200 cm). These maps are given for mean values, but also for $0.05^{th}$, $50^{th}$ and $95^{th}$ percentiles and an uncertainty
estimate. To match the geological attributes described later in this paragraph we also derive porosity and conductivity estimates
from the mean sand and clay values for each layer using the regression equations described by Cosby et al. (1984). However,
SOILGRIDS data are estimated for depths up to 2 meters everywhere, without taking into account the actual depth to bedrock
of any location. Thus, second, we included maps from the Pelletier soil database (Pelletier et al., 2016a, b). These distinguish
between uplands, valley bottoms and lowlands and provide estimates of the depths of soil, intact regolith, and sedimentary
deposits above unweathered bedrock. These variables may be used to set more realistic soil depths in models compared to a
spatially uniform depth. Third, we include cut-outs from the GLHYMPS data (Gleeson et al., 2014; Gleeson, 2018) as poly-
gons. Contained as attributes are estimates of geologic permeability and porosity, which may be used to parametrize models.



**Long-term monthly climate means**
Data: WorldClim
Original resolution: 30 arcsecond
Monthly climate variable means based on 1970-2000 data. Used to derive various values (source data cannot be redistributed).

**Leaf Area Index**
Data: MCD15A2H.061
Original resolution: 500m
LAI estimates for 2002-07-04 to 2023-10-08 at 8-day steps.

**Forest height**
Data: Global Land Cover & Land Use Change
Original resolution: 30m
Estimated vegetation height in 2000 and 2020.

**Agriculture**
Data: LGRIP30v001
Original resolution: 30m
Land use classification into rainfed, irrigated or no agriculture.

**Land cover**
Data: MCD12Q1.061
Original resolution: 500m
Land use classification into 17 broad types such as deciduous broadleaf forest, tundra and urban.

**Land cover**
Data: Global Land Cover & Land Use
Original resolution: 30m
Land use classification distinguishing primarily between cropland, urban, wetlands and solid ground.

**Open water**
Data: HydroLAKES
Original resolution: n/a (polygon)
Various characteristics about lakes and reservoirs.

**Topography**
Data: MERIT Hydro
Original resolution: 3 arcsecond
Surface elevation, slope, aspect.

**N**

**Soil properties**
Data: SOILGRIDS 2.0
Original resolution: 250m
Various soil properties at 6 depths, including uncertainty estimates.

**Soil properties**
Data: Pelletier
Original resolution: 30 arcsecond
Estimates of soil, intact regolith and sedimentary layer depth.

**Geology**
Data: GLHYMPS
Original resolution: n/a (polygon)
Estimates of permeability and porosity.

**Figure 4.** Overview of geospatial maps provided for each catchment in the CAMELS-SPAT data set, using a transboundary basin as an example (Canadian gauge ID: *05AD003*; sub-basin outlines given in black in all data layers apart from topography). The topography layer also shows the basin's gauge location as a red circle, the different sub-basins with white outlines, and the river network and lakes in blue.





## 3 Catchment Attributes

Existing large-sample data sets cover a wide variety of catchment attributes. An informal analysis of some of the CAMELS
data sets listed in Table 1 shows that these data sets together contain close to 300 different attributes, though any given in-
dividual data set contains no more than 50 to slightly over a 100 of those. Overlap between attributes provided by existing
data sets is moderate at best, partly as a consequence of the data products included in each individual data set. This lack of
uniformity is compounded by a lack of unified terminology, where different data sets may use the same terms to describe
different calculations, or different terms to describe the same attribute. This is in line with findings by Tarasova et al. (2023),
who analyze how 742 journal articles describe the hydroclimatic conditions of their study areas. They find that authors use a
wide variety of attributes with only occasional verification of their attributes' usefulness. Relevant for our work, and in line
with a cursory overview of attributes provided by the data sets listed in Table 1, they also find that the existing literature only
rarely uses catchment descriptors that attempt to quantify the range a particular variable may cover in a given catchment (the
CAMELS-SE data set, Teutschbein (2024), is a notable exception).

We thus made a necessarily subjective choice in which attributes to calculate for the CAMELS-SPAT basins. We aimed
for overlap with existing data sets when possible, and to be mindful of the findings of Tarasova et al. (2023). In particular, in
addition to the commonly provided mean attribute values we also selected statistics that describe the range of an attribute's
values. Examples include the minimum, maximum and standard deviation of vegetation height to give an impression of the
spatial variability in the forest height data, and the inclusion of monthly mean forcing variables to give an impression of the
415 climatic seasonality that is only superficially captured by average seasonality attributes commonly found in other data sets. A
list of all 1178 attributes, divided into five main categories: (1) climate; (2) topography and open water; (3) vegetation and land
cover; (4) subsurface; and (5) hydrology, can be found in Tables A1-A11. We calculate the attribute values at both the basin and
the sub-basin level (excepting streamflow statistics, which are only available at the basin outlet). Further details are provided
in the following sub-sections, though for obvious reasons we do not discuss every individual attribute. We focus the following
description of CAMELS-SPAT attributes instead on providing various examples that highlight why the recommendations in
Tarasova et al. (2023) are important.

### 3.1 Climate attributes

The climatic data included in CAMELS-SPAT, time series of meteorological forcing variables from RDRS and monthly maps of
mean climatic conditions from WorldClim, provide a unique opportunity to characterize each catchment's climatic conditions
in time and space. From the RDRS data we are able to determine seasonal variability, and its variance over multiple years.
From the WorldClim data we are able to characterize the seasonal variability and its variance across space. This leads to a
relatively large number of climatic attributes compared to other data sets, and provides some insight in the variability in time
and space of the drivers of hydrologic behaviour.

Tables A1-A4 list the climatic attributes provided with CAMELS-SPAT. These cover annual mean values of variables of
430 interest (such as precipitation, potential evapotranspiration and snow) commonly found in other datasets, as well as standard





deviations for these values. We expand upon existing data sets by also providing monthly means and monthly standard deviations of all forcing variables, to allow more in-depth investigation of each catchment's seasonality. Figure 5 shows why going beyond annual mean values may be important. Figures 5a and 5b show long-term average aridity and the fraction of precipitation falling as snow (determined on a per-timestep basis using a $0^{\circ}$C threshold; see also Section 4.2.6 for some further

discussion about the PET estimates available in CAMELS-SPAT.). The broad geographical patterns seen here are not particularly surprising, but are, importantly, not necessarily representative of climatic variability on a year-to-year basis (Figure 5c, 5d) or of the range of conditions within each catchment (Figure 5e, 5f). For example, across the great plains area and particularly in the southwestern United States the year-to-year variability in aridity (Figure 5c) can be quite large and certain catchments may fluctuate between arid and humid states on annual timescales. The fraction of precipitation falling as snow equally shows large

inter-annual variability (Figure 5d), with standard deviations close to 10% across a large part of the domain. Within-catchment variability of aridity (Figure 5e) seems modest in most cases but is rather large for snowfall (Figure 5f), highlighting why treating these catchments in a more spatially distributed fashion may be helpful.





**Figure 5.** Selection of climate attributes. (a-d) Statistics derived from RDRS data, showing mean and variability in time. (e-f) Statistics derived from WorldClim data, showing variability within each catchment.





## 3.2 Topography and open water attributes

Topography is a critical control on hydrologic behaviour on both the large and small scale. For example, mountains influence precipitation patterns at the large scale, while at the small scale slope angles affect lateral drainage and topographic features can lead to the formation of lakes. Tables A5 and A6 provide an overview of topographic and open water attributes, respectively. These cover various basic catchment descriptors, such as location and area, and various statistics about the topography and resulting drainage network. Figure 6a and 6b show the catchment elevation mean and standard deviation, respectively. As expected, elevation varies strongly throughout the domain, ranging from sea level to well over 3000 m.a.s.l.. Elevation differences within catchments can be very high in mountainous regions, with prime examples being the northwestern United States and southwestern Canada: the within-catchment standard deviations in elevation are close to 500 m here. Statistics that quantify basin slope (not shown for brevity) show similar patterns, showing that within-catchment topographic drivers of hydrologic behaviour can be highly variable. Topographic conditions lead to a certain amount of open water in the CAMELS-SPAT catchments, with lakes larger than 10 ha being more prevalent in the Canadian basins (Figure 6c) than in basins in the United States. Water storage in these can be considerable (Figure 6d). Stream lengths (Figure 6e and 6f) vary considerably based on the drainage area upstream of each gauge, emphasizing a need for within-catchment routing approaches. The examples in Figure 6 are intended to highlight the variability of conditions within catchments and thus emphasize the need to go beyond treating basins as lumped entities. These examples (particularly Figure 6a and 6b, and 6e and 6f) also illustrate that attributes can show high correlations, suggesting that adding more attributes to an analysis will not necessarily increase the useful information by the same amount. Selecting which attribute to incorporate in any analysis must thus be done somewhat carefully (see also Section 4.2.5).



**Figure 6.** Selection of topographic attributes. Open water (c, d) estimates are obtained from the HydroLAKES database which uses a threshold of 10 ha for lake and reservoir identification. (e, f) Stream length statistics are derived by starting at each headwater sub-basin upstream of a given gauge, and tracing the flow path down until the gauge location is reached. From this ensemble of flow path lengths upstream of a given gauge, the mean and standard deviation of stream lengths are calculated.





## 3.3 Land cover attributes

Table A7 provides an overview of vegetation and land cover attributes. Briefly, these cover various statistics about vegetation height during specific years, monthly Leaf Area Index (LAI) catchment mean and standard deviation, as well as per-catchment counts of three different land class products. We refer the reader to the original publications that describe each dataset for further information about the classes included. Figure 7 provides an example of the spatial (Figure 7a, 7b) and temporal (Figure 7c, 7d) variability in vegetation characteristics. As may be expected, there is considerable variation in vegetation height in space, on both the continental and within-catchment scale. Forested areas in particular exhibit large standard deviations in vegetation height (see for example the Pacific Northwest and western Canada). On a seasonal scale, Leaf Area Index exhibits large variability throughout the domain as a consequence of summer and winter patterns. Vegetation is a key control on hydrologic processes like interception and transpiration, and these images show that mean values do not necessarily capture the complex vegetation patterns that may explain spatial and temporal variability in these processes.

## 3.4 Subsurface attributes

Attributes describing each catchment's subsurface characteristics are listed in Tables A8 and A9. Figure 8a and 8b show SOILGRIDS estimated sand content in the top layer of each catchment and the within-catchment standard deviation of this estimate, respectively. Sand content is often combined with clay and silt content estimates to derive soil parameters used in models, such as porosity and drainage rates. Within-catchment standard deviations tend to be around 20% of the estimated sand content, suggesting that within-catchments drainage properties can vary considerably. For a given depth, the SOILGRIDS property of interest (here: sand content) is estimated with a lower bound (Q0.05), median (Q0.50) and mean value, and upper bound (Q0.95). The prediction uncertainty is then calculated as the ratio of the 90% prediction interval (Q0.95-Q0.05) and the median (Q0.50). Prediction uncertainty (Figure 8c) adds more variability to the sand content estimates, though this is somewhat modest compared to within-basin variability of sand content estimates (Figure 8b). The spatial standard deviation of the uncertainty estimates is even smaller: a couple of percent-point difference at most (Figure 8d). This suggests that the prediction intervals for sand content, in this layer at least, are relatively narrow. The main variability occurs within each catchment, further emphasizing that going beyond lumped representations of hydrologic behaviour may be useful. This is further supported by Figure 8e and 8f, showing the estimated thickness of sedimentary deposits and their spatial standard deviation, respectively. There are clear large-scale patterns of the catchment mean values, where plains and flat areas show the thickest layers. Within-catchment variability is particularly large in catchments with sharp topographic relief (compare Figure 6b) showing the difference in soil structure between high, steep mountains and valley bottoms. However, soil properties are difficult to measure and as a result can be highly uncertain. We urge readers to consult the publications describing these data sets to understand how these values were derived, and how they may feed into new work.





**Figure 7.** Selection of vegetation attributes.





 *(figure)*

**Figure 8.** Selection of subsurface attributes. (c-d) Sand content uncertainty is defined as the ratio between the 90-percentile prediction interval and the median prediction.





## 3.5 Hydrologic signatures

Statistics that describe flow regimes, commonly called signatures, are an active area of research (e.g., McMillan, 2021). As an initial start, we provide the same signatures as provided in the original CAMELS data set and expand upon these in a handful of

495 ways: (1) in addition to mean values, we provide standard deviations when applicable; (2) we provide monthly runoff signatures to complement the monthly climate attributes; (3) we expand the no, low and high flow duration signatures to include median, skew and kurtosis values. For the signatures in Table A10, we calculate the signature per year of data first, and then find the mean and standard deviation (if applicable) across years. For the statistics about no, low and high flow periods (Table A11), we instead use all years together and calculate the statistics from this single longer time series.

A subset of these hydrologic signatures is shown in Figure 9. As expected, the signatures show strong relations to the climate attributes in Figure 5a and 5b. Mean discharge (Figure 9a) is particularly high in non-arid areas, and the standard deviation of annual mean discharge (Figure 9b) suggests strong intra-annual variability in the observed runoff at most gauges. The influence of snow processes can be clearly seen in the differences between May and December mean runoff values (Figure 9c, 9d). Low flow duration (Figure 9e; defined as days where discharge is below 20% of the mean discharge for the basin) emphasizes the

seasonality in runoff patterns in most of these these basins. However, these mean values are likely not particularly representative of the duration of low-runoff events. In the majority of basins, the distributions of low flow durations (as well as no flow and high flow durations; not shown for brevity) are positively skewed (Figure 9f). This indicates that these distributions have heavy tails, and that the mean values will be heavily biased by a relatively small number of events. In many basins, the median duration will provide a more representative value of the typical no, low and high flow durations. Almost all recent large-sample data

sets provide mean duration of no, low and high flow events, but the skewness and kurtosis of the underlying distributions are typically not accounted for. This leads to an overestimation of the typical duration of these events, and may hinder classification efforts. We strongly suggest that the shape of the duration distributions is accounted for in further work.



**Figure 9.** Selection of hydrologic signatures.



## 4  Discussion

### 4.1  Recommendations for data providers

#### 4.1.1  Dimension boundary information in publicly available data

In Sections 2.3 and 2.4 we describe the processing of streamflow observations and meteorological data, respectively. One challenge here is determining the representativeness (or validity) of data values in time and space. Data can be instantaneous (i.e., valid at a specific point in time) or time-averaged (i.e., valid over a specific time window), and treating one as the other leads to incorrect estimates of fluxes and thus state changes in the system (see also the derivation of hourly flow values in the Supplementary Materials). The same concern applies to space: values may be representative for a specific point, or averaged over a given region. Accounting for these differences is not always straightforward, in particular because information about the spatial and temporal validity of publicly available data is not always easily available and may require informal inquiries to obtain. This hampers the correct application and interpretation of data, and can lead to easily preventable biases in analyses and modeling efforts.

A simple solution is provided by the NetCDF Climate and Forecast (CF) Metadata Conventions (see Section 7 in Eaton et al., 2023). These conventions describe the specification of bounds for coordinate variables (i.e., dimensions such as latitude, longitude and time) that indicate between which coordinate values a given data value is considered valid. Specific examples for spatial, gridded data can be found in Section 7.1 in Eaton et al. (2023); time bounds are discussed in Examples 7.5 and 7.6. The CF conventions are designed for NetCDF files but the principle of specifying dimension bounds in time and space, between which data values are valid, is widely applicable. We strongly recommend that including these bounds as part of data distributions becomes standard practice.

#### 4.1.2  Sub-daily flow data derivations

Process-based models can be useful for long-term water assessments, provided that they are parametrized well and that the theoretical underpinnings of the model are valid (e.g., Kirchner, 2006; Clark et al., 2016). In the case of process-based models, assessing a model's physical realism requires observations at sub-daily resolution. In CAMELS-SPAT we therefore construct hourly streamflow series from time series of instantaneous streamflow observations that are publicly available. However, the phrase "streamflow observations" (though common) is somewhat misleading: in almost all cases the observations are of water levels and streamflow values are estimated for a given water level with rating curves. Especially at high observation frequencies these water levels may be subject to random fluctuations unrelated to streamflow magnitude (e.g., due to wind or small eddies), which will translate into streamflow estimates affected by this noise. A cleaner approach would be to find the average hourly water level, and estimate the average hourly flow from this through the station's rating curve. Development and maintenance of rating curves is complex however and rating curves tend to change through time (see for example the description of WSC's procedures in Gharari et al., 2023b). Computing robust sub-daily streamflow estimates will be easier at institutional levels (not least because it requires access to the rating curves) and we express the hope that this may become standard practice.





## 4.2   Guidelines for practical use

Here we outline various considerations that may be useful to readers. Our goal with these is to set expectations for data set use, and highlight potential pitfalls that may not be immediately obvious.

### 4.2.1   Selection of time periods

Our aim with CAMELS-SPAT is to facilitate a wide range of studies, and we have therefore provided as much data for each gauge as seemed feasible. In particular, this meant that we only excluded station observations before 1950, because none of the forcing data sets covers this period, and also accepted the fact that not all forcing products are available for the full period for a given gauge. For different purposes, it will thus be necessary to subset the data we provide to shorter time periods. Table 4 provides an overview of the time periods covered by the various data products that may assist in selecting appropriate periods for specific studies.

**Table 4.** Time periods covered by the different data sets included in CAMELS-SPAT. Geospatial data not listed are static products that have no time dimension.

| Streamflow data | Resolution | Start (min) | End (max) | Notes |
|---|---|---|---|---|
| USGS | Hourly | 1956-12-07 | 2023-01-03 | Varies per gauge, see Fig. 3 and Fig. A1 |
| USGS | Daily | 1950-01-01 | 2023-01-02 | Varies per gauge, see Fig. 3 and Fig. A1 |
| WSC | Hourly | 2021-06-01 | 2023-01-02 | Varies per gauge, see Fig. 3 and Fig. A1 |
| WSC | Daily | 1950-01-01 | 2022-12-31 | Varies per gauge, see Fig. 3 and Fig. A1 |
| Forcing data | Resolution | Start | End | |
| RDRS | Hourly | 1980-01-01 | 2018-12-31 | |
| Daymet | Daily | 1980-01-01 | 2023-12-31 | |
| ERA5 | Hourly | 1950-01-01 | 2023-01-03 | |
| EM-Earth | Hourly | 1950-01-01 | 2019-12-31 | |
| Geospatial data | Resolution | Start | End | |
| MODIS LAI | 8-daily | 2002-07-04 | 2023-10-08 | |
| Forest height | 20-yearly | 2000-01-01 | 2020-01-01 | |

### 4.2.2   Utilization of streamflow data quality flags

We retained streamflow observation quality flags provided by the USGS and WSC during processing and stored these in the same NetCDF files as the streamflow observations themselves. These flags indicate conditions affecting the streamflow measurement, such as the presence of river ice, backwater effects, water levels below sensor level, or equipment malfunction.





These conditions suggest that streamflow data at these time steps are inaccurate and this may affect analyses that use these data.
For example, it is known that errors at individual time steps may have disproportionate effects on aggregated efficiency scores that are used in modeling (e.g., Newman et al., 2015; Clark et al., 2021). Excluding streamflow observations from efficiency score calculations based on data quality flags is a possible way to limit the impacts of known erroneous streamflow values.

### 4.2.3  Spatial validity of meteorological forcing data

CAMELS-SPAT contains meteorological data from four different data sets at their original gridded resolution, as well as
averaged at the basin and sub-basin level. During this averaging process we assumed that values provided at specific coordinates are valid for a grid cell around this point. This is a simplistic approach but it is somewhat difficult to justify more elaborate assumptions (such as some form of interpolation), because in reality the change of meteorological variables in space would be dependent on local topography at scales smaller than the typical forcing data grid cell. Interpolation methods may yield more realistic sub-basin and basin averaged values, but it is beyond the scope of this paper to investigate these.

### 4.2.4  Modelling the Prairie Pothole region

Model performance across the United States is known to change regionally, where model performance is at its worst in the drier central regions (e.g., Newman et al., 2015; Towler et al., 2023). In CAMELS-SPAT we compound this problem by including basins from the so-called Prairie Pothole region. This area covers parts of southern Alberta, Saskatchewan, Manitoba, North Dakota, South Dakota, Minnesota and Iowa, and is colloquially known as "the graveyard of hydrological models" (e.g.,
Muhammad et al., 2019; Budhathoki et al., 2020; Ahmed et al., 2023). The landscape in the Prairie Pothole region is relatively young on geological time scales and large parts of it have not yet eroded into traditional river networks. Surface depressions are common and typically not connected to the stream network, except through very slow groundwater drainage and the occasional fill-and-spill event (Hayashi et al., 2016; Clark and Shook, 2022). In the basins we provide as part of the CAMELS-SPAT data, all sub-basins are connected to the stream network. However, surface depressions below the resolution of the MERIT DEM
are common and will affect hydrologic behaviour in these (sub)basins. We recommend that users account for these potholes in their analyses and modeling efforts, possibly through the use of stand-alone models or post-processing tools (e.g., Clark and Shook, 2022), or by adapting existing models with an appropriate landscape module (e.g., Ahmed et al., 2023), or to adjust their expectations about model performance accordingly.

### 4.2.5  Selection and extension of catchment attributes

We derived various catchment attributes for the basins in CAMELS-SPAT for ease of use and comparison with existing data sets. However, the number of attributes included in CAMELS-SPAT is rather high and we encourage others to make a careful selection of which attributes to use in their own work. Attribute values can show considerable correlations, and using larger number of attributes will not necessarily add an equal amount of new information. Larger numbers of attributes will, however, increase computation and analysis times. A more fruitful approach likely relies on defining hypotheses that can be tested with





catchment attributes, and deliberately selecting the right attributes for these tests. If our initial attribute calculations do not offer the right choices, new attributes can easily be derived from the data products included in CAMELS-SPAT. We refer the reader to Tarasova et al. (2023) for a deeper discussion and recommendations on the use of catchment descriptors. We particularly encourage investigations that evaluate the usefulness of our provided attributes for catchment characterization purposes, in line with those recommendations.

### 595 4.2.6 Potential evapotranspiration estimates

In order to facilitate a wide range of modeling studies, CAMELS-SPAT contains a variety of estimates of potential evapotranspiration (PET). These can be used as inputs to certain types of models, and to calculate certain climatic attributes such as a basin's aridity. However, there are multiple ways to estimate PET depending on data availability and purpose (McMahon et al., 2013) and this results in a certain amount of uncertainty in these PET estimates and any values derived from them. Here we
provide a brief overview of the various PET estimates available in CAMELS-SPAT along with a brief assessment that may help users decide which data to use. Table 5 summarizes this overview.

CAMELS-SPAT contains time series of potential evapotranspiration (PET) data directly obtained from ERA5. However, Clerc-Schwarzenbach et al. (2024) point out that PET data obtained from ERA5-Land must be treated carefully and may include severely unrealistic values. Preliminary analysis suggests this applies to PET values obtained from ERA5 too (see
Section 4 in the Supplementary Materials). We have kept the ERA5 PET estimates for users who wish to investigate this further, but urge caution about their use.

CAMELS-SPAT also contains time series of PET estimates obtained with the Penman-Monteith method and hourly RDRS data, as well as time series of PET estimates obtained with the Priestly-Taylor method and daily Daymet data. Finally, we included spatial PET estimates using the temperature-based method in Oudin et al. (2005), applied to monthly averaged World-
610 Clim data. Equations for all three approaches can be found in Section 2.5 in the Supplementary Materials. We compared these to the PET estimates from Singer et al. (2021) and their overview of mean annual PET estimates from various products in their Figure 1 and Table 2. Preliminary analysis (see Section 4 in the Supplementary Materials) suggests that our PET estimates from RDRS, Daymet and WorldClim all exhibit similar spatial patterns as the five data sets shown in Singer et al. (2021). Visual comparison also suggests that there is some spread in the magnitude of our estimates. Monthly estimates based on WorldClim
data are low compared to the other methods and data sources, and comparable to those in GLEAM. Daily estimates based on Daymet data are close to the middle of the range of estimates. Hourly estimates based RDRS data are within the ranges of estimates provided by the other methods and data sets, though somewhat high compared to most other products.

Due to the lack of uniformity in PET definitions and calculation methods (e.g. McMahon et al., 2013), it is difficult to say which estimates are the most accurate. For time series, any expected systematic biases could be corrected before using the time
series as model input. Derived statistics with clear physical interpretations, such as aridity, are more difficult. A basin may be classified as either water-limited or energy-limited solely as a consequence of the data and PET estimation method used, and this may hinder classification and interpretation efforts. Possible ways around this may involve the use of multiple estimates of PET-related attributes. We thus recommend caution when selecting and interpreting any PET estimates for further use.



**Table 5.** Overview of PET estimates in CAMELS-SPAT, their use, and a summary of how these values compare to each other as well as the estimates from five other PET estimates listed in Singer et al. (2021).

| Source data | Temporal resolution | PET estimation method | Used for | Assessment |
|---|---|---|---|---|
| ERA5 | hourly | Unknown | - | Likely incorrect in multiple locations. |
| RDRS | hourly | Penman-Monteith | Climate attributes | Plausible patterns; values somewhat high compared to most estimates. |
| Daymet | daily | Priestly-Taylor | - | Plausible patterns; values close to the middle of all estimates. |
| WorldClim | monthly | Eq. 3 in Oudin et al. (2005) | Climate attributes | Plausible patterns; values on the lower end of all estimates. |

### 4.3 Potential improvements

CAMELS-SPAT represents a substantial data processing effort, but further enhancements are possible. We briefly list these here. First, approximately 15% of our basin outlines have been assigned confidence ratings of medium or low. Future efforts can focus on refining these outlines, through further manual intervention, or higher resolution DEMs, or both. Second, we were somewhat limited in our ability to obtain hourly streamflow observations for the Canadian basins. Extension of these records would be helpful. Third, we necessarily needed to limit the extent of our geographical domain and this means there

is a limit to the different types of landscapes our data set covers. However, apart from Daymet and RDRS, all data sets used here have global coverage. Combination with local streamflow observations, and possibly high-quality local data sets, should allow for straightforward extension of the data set to other regions. The code available on our GitHub repository could provide a starting point for such efforts. Fourth, extending the dataset to include observations or estimates of variables of interest other than streamflow would help with multi-variate analysis and model evaluation. Examples include satellite observations of snow

cover, or estimates of evaporation fluxes or water storage in the soil.

### 4.4 Data set structure and size

For convenience, we divided the collection of 1426 CAMELS-SPAT gauges into various subsets. At the highest level, we structured the data set with different folders for attributes, forcing data, geospatial data, observations and shapefiles. At the next level, we divided the data set into three categories of headwater, meso-scale and macro-scale basins. Headwater basins are

640 defined as catchments with only a single sub-basin in our delineation. Meso-scale basins are basins that are not headwaters and below a total area of $10^3$ km$^2$, and macro-scale basins are those with areas between $10^3$ km$^2$ and $10^4$ km$^2$. Headwater basins account for 304 out of 1426 total (mean area of approximately 60 km$^2$), 727 basins fall in our meso-scale category (mean area $\approx 400$ km$^2$, with on average 9 sub-basins), and the remaining 446 basins are macro-scale basins (mean area $\approx 3000$ km$^2$, on average 66 sub-basins). From here we divided the data set into further subfolders when convenient.



The total size of the CAMELS-SPAT data is approximately 5.5TB. Almost all of this is forcing data (5.4TB) and specifically the gridded variants of the forcing data (4.3TB). Basin-averaged data (summed for all four forcing data sets) sums up to 85GB, while distributed forcing data (i.e., averaged at the sub-basin level) sums up to not quite 1.2TB. A full overview of the size of various components of the dataset can be found on the data repository. This overview, combined with the overall folder structure should allow users to fine-tune their downloads easily. Further instructions to include or exclude components from

the download can be found on the data repository.

## 5   Conclusions

This paper describes the development of the CAMELS-SPAT data set. Our goal is to enable a wide range of hydrologic studies, with a particular focus on hydrologic modeling, by performing a wide range of data processing steps and sharing both the code and outcomes of these. We extend the original CAMELS data (Newman et al., 2015; Addor et al., 2017a) in four ways to

achieve this goal. First, we extend the geographical domain of the data set beyond the contiguous United States by including Canadian basins. Second, we provided meteorological data specifically aimed at spatially-distributed physics-based hydrologic models, in addition to the inputs needed to run lumped, conceptual models. Third, we provide maps of multiple geospatial data sets for each basin, rather than only a selection of summary statistics derived from these maps. Fourth, we provide a variety of catchment attributes intended to describe the spatial and temporal range of our attributes, in addition to the more commonly

provided mean attribute values.

CAMELS-SPAT thus consists of meteorological data, streamflow observations and geospatial data for 1426 basins across the United States and Canada. The meteorological data includes a number of variables typically associated with process-based models, as well as potential evapotranspiration estimates that can be used with the more conceptual model types, at hourly time steps. This forcing data is provided in gridded format at its own resolution, as well as spatially averaged at the sub-basin and

basin level. Streamflow observations are provided at daily time steps and complemented with hourly observations when these are available. Geospatial data, covering vegetation, land use, topography, soil and geology, are provided as geo-referenced maps for each basin, from which model inputs or summary statistics that go beyond our provided attributes can easily be derived. Finally, the information for each gauge (streamflow, meteorological, geospatial data) are summarized in an extensive number of catchment attributes, at both the basin and sub-basin level.

In developing CAMELS-SPAT, we focused on providing the necessary data for a wide variety of studies. We envision the data being helpful for a variety of studies aimed at improving our understanding of hydrologic processes and our ability to model those processes. By removing the need for a considerable amount of cumbersome data processing, we hope CAMELS-SPAT can support a wide range of hydrologic investigations at a fraction of the effort otherwise needed.

The data set can be accessed through the Federated Research Data Repository (FRDR) through: https://doi.org/10.20383/1

03.01216. When using CAMELS-SPAT, please note the attribution and license requirements for data set components outlined Section 6.



## 6 Code and data availability

The complete CAMELS-SPAT data set can be accessed through the Federated Research Data Repository (FRDR) through: https://doi.org/10.20383/103.01216. Code needed to reproduce CAMELS-SPAT data preparation is available on GitHub: https://github.com/ch-earth/camels_spat. Data source used in the preparation of this manuscript are listed below, separated into data used but not redistributed and data that is redistributed. These data products are provided under a variety of licenses. Please see the individual licenses for detail, and note that attribution is in almost all cases mandatory. We have provided a **data_citation.bib** file available on the CAMELS-SPAT data repository and ask users to cite each separate data set that we redistribute in any publications that use CAMELS-SPAT. Elements in CAMELS-SPAT not covered below (processing code, attributes) are provided under a CC-BY-NC 4.0 license.

### 6.1 Data (redistributed)

Listed here are details about each of the data sets used in the creation of and partly reproduced in the CAMELS-SPAT data.

#### 6.1.1 Meteorological data

Meteorological forcing fields were obtained from the Daymet v4.1 data set (Thornton et al., 2021, 2022), which is openly shared, without restriction, in accordance with the NASA Earth Science Data and Information System (ESDIS) Project Data Use Policy. For license terms, see https://www.earthdata.nasa.gov/learn/use-data/data-use-policy (accessed: 2024-05-24).

Meteorological forcing fields were obtained from the ERA5 data set (Hersbach et al., 2020, 2017, 2023) under the Copernicus Data License (https://cds.climate.copernicus.eu/cdsapp#!/home). For license terms, see: https://cds.climate.copernicus.eu/api/v2/terms/static/licence-to-use-copernicus-products.pdf (accessed: 2023-12-18). Redistributed ERA5 data were generated using Copernicus Climate Change Service information [2023] in the case of the gridded forcing files. CAMELS-SPAT also contains modified Copernicus Climate Change Service information [2023] in the case of the (sub)basin-averaged forcing files. Neither the European Commission nor ECMWF is responsible for any use that may be made of the Copernicus information or data it contains.

Meteorological forcing fields were obtained from the Deterministic EM-Earth data set Tang et al. (2022a, b) under a CC-BY 4.0 license (https://www.frdr-dfdr.ca/repo/dataset/8d30ab02-f2bd-4d05-ae43-11f4a387e5ad).

Meteorological forcing fields were obtained from the RDRS v2.1 data set (Gasset et al., 2021, data source: Environment and Climate Change Canada) under the Environment and Climate Change Canada Data Server End-Use Licence version 2.1. For license terms, see: https://eccc-msc.github.io/open-data/licence/readme_en/ (accessed 2025-02-07).

#### 6.1.2 Basin outlines

Sub-basin polygons were obtained from the MERIT Basins data set (Lin et al., 2019, http://hydrology.princeton.edu/data/mpan/MERIT_Basins/). No formal license is stated in the paper, but data has since been moved elsewhere (https://www.reachhydro.org/home/params/merit-basins, last access: 2025-02-07) and is available there under a CC-BY-NC-SA 4.0 license.



Reference shapefiles for the basins in the United States were obtained from the CAMELS data set (Newman et al., 2015; Addor et al., 2017a, https://doi.org/10.5065/D6MW2F4D). The source of these shapefiles if the U.S. Geological Survey
HCDN-2009 data set (Lins, 2012), and as such considered to be in the public domain (see: https://www.usgs.gov/information -policies-and-instructions/copyrights-and-credits, [last access: 2024-03-21]).

The first set of reference shapefiles for the basins in Canada were obtained from the National hydrometric network basin polygons data set (Environment and Climate Change Canada, 2020b, https://open.canada.ca/data/en/dataset/0c121878-ac23-4 6f5-95df-eb9960753375), available under the Open Government License - Canada (https://open.canada.ca/en/open-governm
ent-licence-canada, [last access: 2024-03-21]).

The second set of reference shapefiles for the basins in Canada were obtained from the Reference Hydrometric Basin Network (Government of Canada, 2022, https://www.canada.ca/en/environment-climate-change/services/water-overview/ quantity/monitoring/survey/data-products-services/reference-hydrometric-basin-network.html), available under unknown license.

### 6.1.3   Streamflow data

Daily flow data for the basins in the United States were obtained from the Daily Values Service, courtesy of the U.S. Geological Survey (https://nwis.waterservices.usgs.gov/docs/dv-service/daily-values-service-details/, [last access: 2024-03-21]). Data are considered to be in the public domain (see: https://www.usgs.gov/information-policies-and-instructions/copyrights-and-c redits, [last access: 2024-03-21])

Hourly flow data for the basins in the United States were derived from the high-resolution Instantaneous Values Service (source: U.S. Geological Survey, https://nwis.waterservices.usgs.gov/docs/instantaneous-values/instantaneous-values-details/, [last access: 2024-03-21]). Data are considered to be in the public domain (see: https://www.usgs.gov/information-policies-a nd-instructions/copyrights-and-credits, [last access: 2024-03-21]).

Daily flow data for the basins in Canada were obtained from the HYDAT database version *20230505*, courtesy of the Water
Survey of Canada (https://www.canada.ca/en/environment-climate-change/services/water-overview/quantity/monitoring/su rvey/data-products-services/national-archive-hydat.html, [last access: 2024-03-21]). Data are considered public information (see: https://wateroffice.ec.gc.ca/disclaimer_info_e.html for full terms and details, [last access: 2024-03-21]). Note that the HYDAT database gets continuously updated, and superseded versions are not publicly available.

Hourly flow data for the basins in Canada were derived from the high-resolution data available on the Web Service Links
(source: Water Survey of Canada, https://wateroffice.ec.gc.ca/services/links_e.html, [last access: 2024-03-21]). Data are considered public information (see: https://wateroffice.ec.gc.ca/disclaimer_info_e.html for full terms and details, [last access: 2024-03-21]).

### 6.1.4   Geospatial data

Forest height grids were obtained from the Global Land Cover and Land Use Change, 2000-2020 data set (Potapov et al., 2021)
under a CC-BY license (https://glad.umd.edu/dataset/GLCLUC2020/).



Leaf Area Index grids were obtained from the MCD15A2H.061 data set (Myneni et al., 2021, https://lpdaac.usgs.gov/produc ts/mcd15a2hv061/). Data can be redistributed with no restriction. See: https://lpdaac.usgs.gov/data/data-citation-and-policies/ (accessed: 2023-10-17).

Agriculture grids were obtained from the LGRIP30 data set (Thenkabail et al., 2021; Teluguntla et al., 2023, https://lpdaac .usgs.gov/products/lgrip30v001/). Data can be redistributed with no restriction. See: https://lpdaac.usgs.gov/data/data-citatio n-and-policies/ (accessed: 2023-10-17).

Land cover and land use grids were obtained from the MCD12Q1.061 data set (Friedl and Sulla-Menashe, 2022, https: //lpdaac.usgs.gov/products/mcd12q1v061/). Data can be redistributed with no restriction. See: https://lpdaac.usgs.gov/data/dat a-citation-and-policies/ (accessed: 2023-10-17).

Land cover and land use grids were obtained from the Global land cover and land use 2019 data set (Hansen et al., 2022) under a CC-BY 4.0 license (https://glad.umd.edu/dataset/global-land-cover-land-use-v1).

Lakes polygons were obtained from the HydroLAKES data set (Messager et al., 2016) under a CC-BY 4.0 license (https: //www.hydrosheds.org/products/hydrolakes).

Digital Elevation Model grids were obtained from the Merit Hydro Adjusted Elevations data set (Yamazaki et al., 2019)
under CC-BY-NC 4.0 or ODbL 1.0 licenses (http://hydro.iis.u-tokyo.ac.jp/~yamadai/MERIT_Hydro/).

Soil property grids were obtained from the SOILGRIDS 2.0 data set (Poggio et al., 2021) under a CC-BY-NC 4.0 license (https://soilgrids.org/).

Soil property grids were obtained from the Pelletier data set (Pelletier et al., 2016b, a, https://daac.ornl.gov/SOILS/guides /Global_Soil_Regolith_Sediment.html). Data can be redistributed with no restriction. See: https://www.earthdata.nasa.gov/lea
rn/use-data/data-use-policy (accessed: 2023-12-18).

Geology polygons were obtained from the GLHYMPS data set (Gleeson et al., 2014; Gleeson, 2018) under a CC-BY 4.0 license (https://borealisdata.ca/dataset.xhtml?persistentId=doi:10.5683/SP2/DLGXYO).

## 6.2   Data (not redistributed)

Listed here are details about each of the data sets used in the creation of, but not distributed as part of, the CAMELS-SPAT
data.

### 6.2.1   Basin delineation

Flow direction grids were obtained from the Merit Hydro Adjusted Elevations data set (Yamazaki et al., 2019) under CC-BY-NC 4.0 or ODbL 1.0 licenses (http://hydro.iis.u-tokyo.ac.jp/~yamadai/MERIT_Hydro/).

Flow accumulation grids were obtained from the Merit Hydro Adjusted Elevations data set (Yamazaki et al., 2019) under
CC-BY-NC 4.0 or ODbL 1.0 license (http://hydro.iis.u-tokyo.ac.jp/~yamadai/MERIT_Hydro/).





### 6.2.2 Geospatial data

Climate grids were obtained from the WorldClim data set (Fick and Hijmans, 2017, https://www.worldclim.org/data/worldclim21.html). WorldClim data were used to calculate high-resolution climate attributes and derive a number of maps. The source data cannot be redistributed.

**Appendix A: Streamflow data availability**

Figure A1 shows streamflow data availability at a more granular level than the aggregated data in Figure 3.





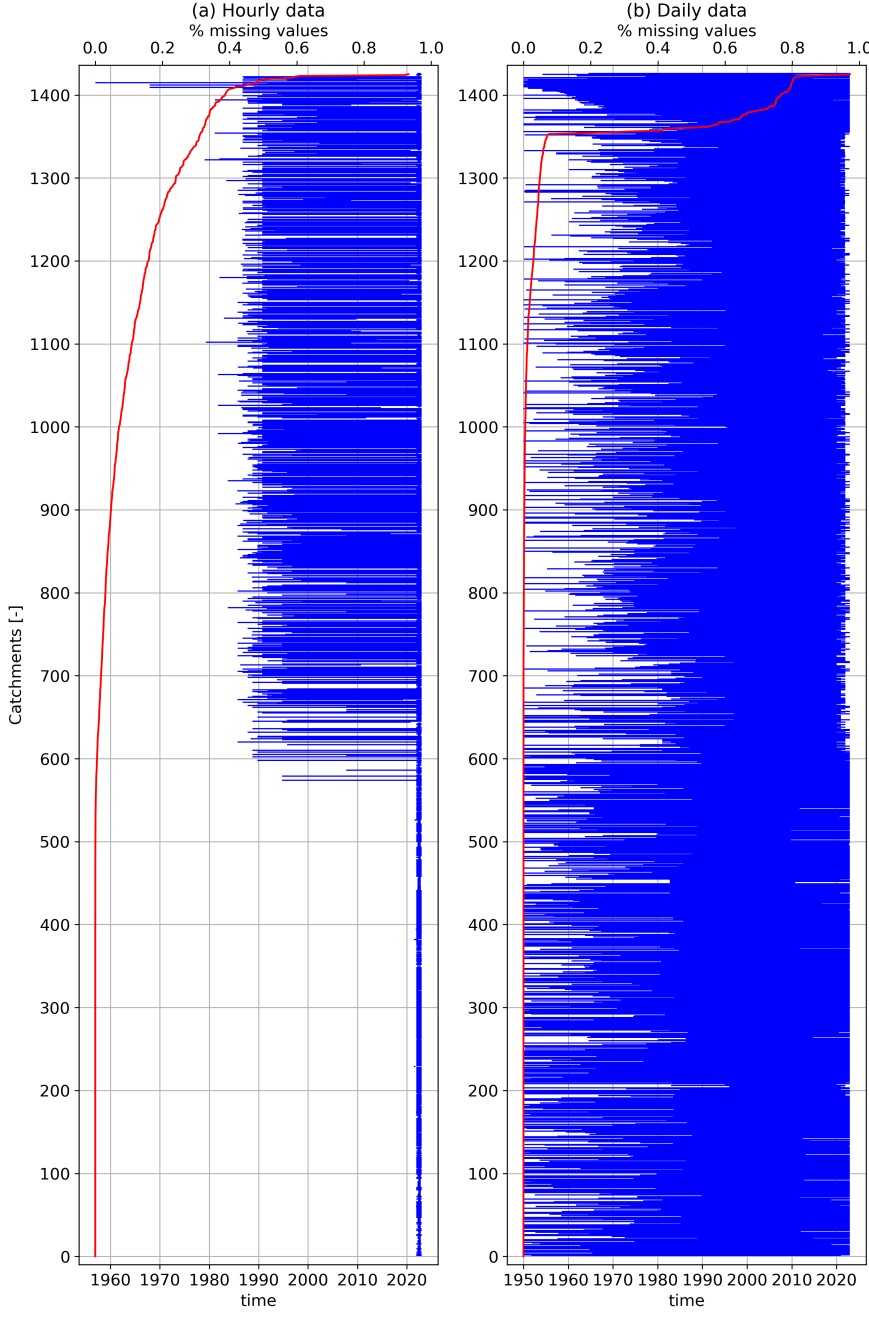

**Figure A1.** Flow data availability for gauges included in CAMELS-SPAT. The period on the lower x-axis refers to the period between the first publicly available flow record for a given station and its last, with this record period given in blue for each gauge. Missing values occur within this record period and are given here as percentages in red on the top x-axis.





**Table A1.** Climate attributes: annual statistics.

| Attribute | Description | Units | Data source |
|---|---|---|---|
| num_years_rdrs | Number of years of RDRS data used to calculate attributes | years | RDRS |
| PR0_mean | Mean annual average precipitation total | mm | RDRS[1] |
| PR0_std | Standard deviation of annual average precipitation total | mm | RDRS[1] |
| prec_mean | Mean annual average precipitation total | mm | WorldClim |
| prec_std | Standard deviation of annual average precipitation total | mm | WorldClim |
| pet1_mean | Mean annual average potential evapotranspiration (PET) total | mm | RDRS[1] |
| pet1_std | Standard deviation of annual average PET total | mm | RDRS[1] |
| pet2_mean | Mean annual average potential evapotranspiration (PET) total | mm | WorldClim[2] |
| pet2_std | Standard deviation of annual average PET total | mm | WorldClim[2] |
| TT_mean | Mean of annual mean daily average temperature | °C | RDRS[1] |
| TT_std | Standard deviation of annual mean daily average temperature | °C | RDRS[1] |
| tavg_mean | Mean annual average temperature | °C | WorldClim |
| tavg_std | Spatial standard deviation of annual average temperature | °C | WorldClim |
| aridity1_mean | Mean annual aridity (PET/P) | − | RDRS |
| aridity1_std | Standard deviation of annual aridity (PET/P) | − | RDRS |
| aridity2_mean | Mean annual aridity (PET/P) | − | WorldClim |
| aridity2_std | Standard deviation of annual aridity (PET/P) | − | WorldClim |
| seasonality1_mean | Mean precipitation seasonality compared to temperature seasonality[4] | − | RDRS |
| seasonality1_std | Standard deviation of precipitation seasonality compared to temperature seasonality[4] | − | RDRS |
| seasonality2_mean | Mean precipitation seasonality compared to temperature seasonality[5] | − | WorldClim |
| seasonality2_std | Standard deviation of precipitation seasonality compared to temperature seasonality[5] | − | WorldClim |
| fracsnow1_mean | Mean annual snow fraction (°C degree threshold) | − | RDRS |
| fracsnow1_std | Standard deviation of annual snow fraction (°C degree threshold) | − | RDRS |
| fracsnow2_mean | Mean annual snow fraction (°C degree threshold) | − | WorldClim |
| fracsnow2_std | Standard deviation of annual snow fraction (°C degree threshold) | − | WorldClim |

[1] For consistency, we converted the RDRS units into those used in WorldClim.

[2] Computed using WorldClim's *srad* and *tavg* variables, and Equation 3 in Oudin et al. (2005).

[3] For consistency, we converted the WorldClim units into those used in RDRS.

[4] Calculated using Eq.14 in Woods (2009) for daily data from individual years, then finding the mean and standard deviation across years.

[5] Calculated using Eq.14 in Woods (2009) using monthly data; i.e. a much coarser temporal resolution then RDRS.





**Table A2.** Climate attributes - continued: frequency, duration and timing of high and low precipitation, and high and low temperature periods.

| Attribute | Description | Units | Data source |
|---|---|---|---|
| low_temp_freq | Frequency of cold days ($< 0^o$C) | days year$^{-1}$ | RDRS |
| low_temp_dur_mean | Mean duration of cold days ($< 0^o$C) | days | RDRS |
| low_temp_dur_median | Median duration of cold days ($< 0^o$C) | days | RDRS |
| low_temp_dur_skew | Skew of cold day durations ($< ^o$C) | – | RDRS |
| low_temp_dur_kurtosis | Kurtosis of cold day durations ($< 0^o$C) | – | RDRS |
| low_temp_timing | Season during which most cold days occur ($< ^o$C) | season | RDRS |
| high_temp_freq | Frequency of hot days ($>$ mean daily max $+5^o$C)[1] | days year$^{-1}$ | RDRS |
| high_temp_dur_mean | Mean duration of hot days ($>$ mean daily max $+5^o$C) | days | RDRS |
| high_temp_dur_median | Median duration of hot days ($>$ mean daily max $+5^o$C) | days | RDRS |
| high_temp_dur_skew | Skew of hot day durations ($>$ mean daily max $+5^o$C) | – | RDRS |
| high_temp_dur_kurtosis | Kurtosis of hot day durations ($>$ mean daily max $+5^o$C) | – | RDRS |
| high_temp_timing | Season during which most hot days occur ($>$ mean daily max $+5^o$C) | season | RDRS |
| low_prec_freq | Frequency of dry[2] days ($< 1$ mm day$^{-1}$) | days year$^{-1}$ | RDRS |
| low_prec_dur_mean | Mean duration of dry days ($< 1$ mm day$^{-1}$) | days | RDRS |
| low_prec_dur_median | Median duration of dry days ($< 1$ mm day$^{-1}$) | days | RDRS |
| low_prec_dur_skew | Skew of dry day durations($< 1$ mm day$^{-1}$) | – | RDRS |
| low_prec_dur_kurtosis | Kurtosis of dry day durations ($< 1$ mm day$^{-1}$) | – | RDRS |
| low_prec_timing | Season during which most dry days occur ($< 1$ mm day$^{-1}$) | season | RDRS |
| high_prec_freq | Frequency of wet[2] days ($\geq 5$ times mean daily precipitation) | days year$^{-1}$ | RDRS |
| high_prec_dur_mean | Mean duration of wet days ($\geq 5$ times mean daily precipitation) | days | RDRS |
| high_prec_dur_median | Median duration of wet days ($\geq 5$ times mean daily precipitation) | days | RDRS |
| high_prec_dur_skew | Skew of wet day durations ($\geq 5$ times mean daily precipitation) | – | RDRS |
| high_prec_dur_kurtosis | Kurtosis of wet day durations ($\geq 5$ times mean daily precipitation) | – | RDRS |
| high_prec_timing | Season during which most wet days occur ($\geq 5$ times mean daily precipitation) | season | RDRS |

[1] Derived from the World Meteorological Organization's definition of heat waves: a 5-day or longer period with maximum daily temperatures $5^o$C above the "standard" daily maximum temperature. Standard is defined as the mean daily max on each day, using the period 1961-1990 as base. Here we define a hot day as a day where the maximum temperature is at least $5^o$C over the long-term daily maximum temperature. We do not have data for the period 1961-1990 for all basins, and therefore use all data available for a given basin to find the long-term daily maximum temperatures.

[2] For consistency, we use the same definitions of dry and wet days as used in Addor et al. (2017a).





**Table A3.** Climate attributes - continued: spatial and temporal variability in climatic conditions. Attributes ending in _*{X}* are calculated per month, with X ranging from *01* to *12*. Statistics derived from RDRS are calculated over time; statistics derived from WorldClim are calculated across space.

| Attribute | Description | Units | Data source |
|---|---|---|---|
| PR0_mean_month_{X} | Mean monthly average precipitation total | mm | RDRS[1] |
| PR0_std_month_{X} | Standard deviation of monthly average precipitation total | mm | RDRS[1] |
| prec_mean_month_{X} | Mean monthly average precipitation total | mm | WorldClim |
| prec_std_month_{X} | Standard deviation of monthly average precipitation total | mm | WorldClim |
| pet1_mean_month_{X} | Mean monthly average potential evapotranspiration (PET) total | mm | RDRS[1] |
| pet1_std_time_month_{X} | Standard deviation of monthly average PET total | mm | RDRS[1] |
| pet2_mean_month_{X} | Mean monthly average potential evapotranspiration (PET) total | mm | WorldClim[2] |
| pet2_std_month_{X} | Standard deviation of monthly average PET total | mm | WorldClim[2] |
| tdavg_mean_month_{X} | Mean of monthly mean daily average temperature | °C | RDRS[1] |
| tdavg_std_month_{X} | Standard deviation of monthly mean daily average temperature | °C | RDRS[1] |
| tavg_mean_month_{X} | Mean monthly average temperature | °C | WorldClim |
| tavg_std_month_{X} | Spatial standard deviation of monthly average temperature | °C | WorldClim |
| tdmin_mean_month_{X} | Mean of monthly mean daily minimum temperature | °C | RDRS[1] |
| tdmin_std_time_month_{X} | Standard deviation of monthly mean daily minimum temperature | °C | RDRS[1] |
| tmin_mean_month_{X} | Mean monthly minimum temperature | °C | WorldClim |
| tmin_std_month_{X} | Standard deviation of monthly minimum temperature | °C | WorldClim |
| tdmax_mean_month_{X} | Mean of monthly mean daily maximum temperature | °C | RDRS[1] |
| tdmax_std_month_{X} | Standard deviation of monthly mean daily maximum temperature | °C | RDRS[1] |
| tmax_mean_month_{X} | Mean monthly maximum temperature | °C | WorldClim |
| tmax_std_month_{X} | Standard deviation of monthly maximum temperature | °C | WorldClim |
| FB_mean_month_{X} | Mean monthly downward shortwave radiation | $\mathrm{W\,m^{-2}}$ | RDRS |
| FB_std_month_{X} | Standard deviation of monthly downward shortwave radiation | $\mathrm{W\,m^{-2}}$ | RDRS |
| srad_mean_month_{X} | Mean monthly downward shortwave radiation | $\mathrm{W\,m^{-2}}$ | WorldClim[3] |
| srad_std_month_{X} | Standard deviation of monthly downward shortwave radiation | $\mathrm{W\,m^{-2}}$ | WorldClim[3] |

[1] For consistency, we converted the RDRS units into those used in WorldClim.

[2] Computed using WorldClim's *srad* and *tavg* variables, and Equation 3 in Oudin et al. (2005).

[3] For consistency, we converted the WorldClim units into those used in RDRS.



**Table A4.** Climate attributes - continued: spatial and temporal variability in climatic conditions. Attributes ending in _{X} are calculated per month, with X ranging from *01* to *12*. Statistics derived from ERA5 are calculated over time; statistics derived from WorldClim are calculated across space.

| Attribute | Description | Units | Data source |
|---|---|---|---|
| FI_mean_month_{X} | Mean monthly downward longwave radiation | $\mathrm{W\,m^{-2}}$ | RDRS |
| FI_std_month_{X} | Standard deviation of monthly downward longwave radiation | $\mathrm{W\,m^{-2}}$ | RDRS |
| P0_mean_month_{X} | Mean monthly surface pressure | kPa | RDRS[1] |
| P0_std_month_{X} | Standard deviation of monthly surface pressure | kPa | RDRS[1] |
| vapr_mean_month_{X} | Mean monthly vapor pressure | kPa | WorldClim |
| vapr_std_month_{X} | Standard deviation of monthly vapor pressure | kPa | WorldClim |
| HU_mean_month_{X} | Mean monthly specific humidity | $\mathrm{kg\,kg^{-1}}$ | RDRS |
| HU_std_month_{X} | Standard deviation of monthly specific humidity | $\mathrm{kg\,kg^{-1}}$ | RDRS |
| HR_mean_month_{X} | Mean monthly relative humidity | $\mathrm{kPa\,kPa^{-1}}$ | RDRS |
| HR_std_month_{X} | Standard deviation of monthly relative humidity | $\mathrm{kPa\,kPa^{-1}}$ | RDRS |
| UVC_mean_month_{X} | Mean monthly wind speed | $\mathrm{m\,s^{-1}}$ | RDRS |
| UVC_std_month_{X} | Standard deviation of monthly wind speed | $\mathrm{m\,s^{-1}}$ | RDRS |
| wind_mean_month_{X} | Mean monthly wind speed | $\mathrm{m\,s^{-1}}$ | WorldClim |
| wind_std_month_{X} | Standard deviation of monthly wind speed | $\mathrm{m\,s^{-1}}$ | WorldClim |
| phi_mean_month_{X} | Circular mean monthly wind direction | ° | RDRS |
| phi_std_month_{X} | Circular standard deviation of monthly wind direction | ° | RDRS |
| aridity1_mean_month_{X} | Mean monthly aridity (PET/P) | − | RDRS |
| aridity1_std_month_{X} | Standard deviation of monthly aridity | − | RDRS |
| aridity2_mean_month_{X} | Mean monthly aridity (PET/P) | − | WorldClim |
| aridity2_std_month_{X} | Standard deviation of monthly aridity | − | WorldClim |
| fracsnow1_mean_month_{X} | Mean monthly snow fraction (°C degree threshold) | − | RDRS |
| fracsnow1_std_month_{X} | Standard deviation of monthly snow fraction | − | RDRS |
| fracsnow2_mean_month_{X} | Mean monthly snow fraction (°C degree threshold) | − | WorldClim |
| fracsnow2_std_month_{X} | Standard deviation of monthly snow fraction | − | WorldClim |

[1] For consistency, we converted the RDRS units into those used in WorldClim.





**Table A5.** Topographic attributes.

| Attribute | Description | Units | Data source |
|---|---|---|---|
| centroid_lat | Basin centroid latitude | degrees | Varies |
| centroid_lon | Basin centroid longitude | degrees | Varies |
| gauge_lat | Station latitude | degrees | Varies |
| gauge_lon | Station longitude | degrees | Varies |
| basin_area | Basin area | $km^2$ | MERIT Hydro |
| elev_min | Minimum elevation | m.a.s.l. | MERIT Hydro |
| elev_mean | Mean elevation | m.a.s.l. | MERIT Hydro |
| elev_max | Maximum elevation | m.a.s.l. | MERIT Hydro |
| elev_std | Standard deviation of elevation | m.a.s.l. | MERIT Hydro |
| slope_min | Minimum slope | degrees[1] | MERIT Hydro |
| slope_mean | Mean slope | degrees | MERIT Hydro |
| slope_max | Maximum slope | degrees | MERIT Hydro |
| slope_std | Standard deviation of slope | degrees | MERIT Hydro |
| aspect_min | Minimum aspect | degrees[2] | MERIT Hydro |
| aspect_mean | Mean aspect | degrees | MERIT Hydro |
| aspect_max | Maximum aspect | degrees | MERIT Hydro |
| aspect_std | Standard deviation of aspect | degrees | MERIT Hydro |
| stream_length_min | Minimum length from headwater to gauge | km | MERIT Hydro Basins |
| stream_length_mean | Mean length from headwaters to gauge | km | MERIT Hydro Basins |
| stream_length_max | Maximum length from headwater to gauge | km | MERIT Hydro Basins |
| stream_length_std | Standard deviation of length from headwaters to gauge | km | MERIT Hydro Basins |
| stream_length_total | Total stream length | km | MERIT Hydro Basins |
| stream_order_max | Stream order at gauge | − | MERIT Hydro Basins |
| stream_density | Ratio of total stream length and area | $km^{-1}$ | Derived |
| elongation_ratio | Ratio of diameter of circle with same size as basin and longest stream | − | Derived |

[1] Slope angle.

[2] Azimuth that slopes are facing, with 0° indicating North-facing slopes, 90° means East-facing, 180° South-facing, and 270° West-facing.





**Table A6.** Open water attributes. For basins with no identified open water bodies or reservoirs, these attributes will be 0 and NaN.

| Attribute | Description | Units | Data source |
|---|---|---|---|
| open_water_number | Number of open water bodies larger than 10 ha | – | HydroLAKES |
| known_reservoirs | Number of water bodies identified as reservoirs | – | HydroLAKES |
| open_water_area_min | Minimum open water area | $km^2$ | HydroLAKES |
| open_water_area_mean | Mean open water area | $km^2$ | HydroLAKES |
| open_water_area_max | Maximum open water area | $km^2$ | HydroLAKES |
| open_water_area_std | Standard deviation of open water area | $km^2$ | HydroLAKES |
| open_water_area_total | Total open water area | $km^2$ | HydroLAKES |
| open_water_volume_min | Minimum open water volume | $km^2$ | HydroLAKES |
| open_water_volume_mean | Mean open water volume | $km^2$ | HydroLAKES |
| open_water_volume_max | Maximum open water volume | $km^2$ | HydroLAKES |
| open_water_volume_std | Standard deviation of open water volume | $km^2$ | HydroLAKES |
| open_water_volume_total | Total open water volume | $km^2$ | HydroLAKES |
| reservoir_area_min | Minimum reservoir area | $km^2$ | HydroLAKES |
| reservoir_area_mean | Mean reservoir area | $km^2$ | HydroLAKES |
| reservoir_area_max | Maximum reservoir area | $km^2$ | HydroLAKES |
| reservoir_area_std | Standard deviation of reservoir area | $km^2$ | HydroLAKES |
| reservoir_area_total | Total reservoir area | $km^2$ | HydroLAKES |
| reservoir_volume_min | Minimum reservoir volume | $km^2$ | HydroLAKES |
| reservoir_volume_mean | Mean reservoir volume | $km^2$ | HydroLAKES |
| reservoir_volume_max | Maximum reservoir volume | $km^2$ | HydroLAKES |
| reservoir_volume_std | Standard deviation of reservoir volume | $km^2$ | HydroLAKES |
| reservoir_volume_total | Total reservoir volume | $km^2$ | HydroLAKES |



**Table A7.** Vegetation and land cover attributes. Attributes ending in _{X} are calculated per month, with X ranging from *01* to *12*. Attributes ending in _{Y} are calculated for specific years. Attributes ending in _{Z} are categorical attributes, where Z varies between different data sets.

| Attribute | Description | Units | Data source |
|-----------|-------------|-------|-------------|
| lai_mean_month_{X} | Mean monthly Leaf Area Index | $\mathrm{m^2\,m^{-2}}$ | MCD15A2H.061 |
| lai_std_month_{X} | Standard deviation of monthly Leaf Area Index | $\mathrm{m^2\,m^{-2}}$ | MCD15A2H.061 |
| forest_height_{Y}_min | Minimum forest height in year 2000/2020 | m | GLCLUC 2000-2020 |
| forest_height_{Y}_mean | Mean forest height in year 2000/2020 | m | GLCLUC 2000-2020 |
| forest_height_{Y}_max | Maximum forest height in year 2000/2020 | m | GLCLUC 2000-2020 |
| forest_height_{Y}_std | Standard deviation of forest height in year 2000/2020 | m | GLCLUC 2000-2020 |
| lc1_{Z}_fraction | Fraction of land cover class present in the basin | − | GLCLU 2019 |
| lc2_{Z}_fraction | Fraction of land cover class present in the basin | − | MCD12Q1.061 |
| lc3_{Z}_fraction | Fraction of land cover class present in the basin | − | LGRIP30 |





**Table A8.** Subsurface attributes.

| Attribute | Description | Units | Data source |
|---|---|---|---|
| regolith_thickness_min | Minimum upland and hillslope regolith thickness | m | Pelletier[1] |
| regolith_thickness_mean | Mean upland and hillslope regolith thickness | m | Pelletier |
| regolith_thickness_max | Maximum upland and hillslope regolith thickness | m | Pelletier |
| regolith_thickness_std | Standard deviation of upland and hillslope regolith thickness | m | Pelletier |
| soil_thickness_min | Minimum upland and hillslope soil thickness | m | Pelletier |
| soil_thickness_mean | Mean upland and hillslope soil thickness | m | Pelletier |
| soil_thickness_max | Maximum upland and hillslope soil thickness | m | Pelletier |
| soil_thickness_std | Standard deviation of upland and soil regolith thickness | m | Pelletier |
| sedimentary_thickness_min | Minimum upland, valley bottom and lowland sedimentary deposit thickness | m | Pelletier |
| sedimentary_thickness_mean | Mean upland, valley bottom and lowland sedimentary deposit thickness | m | Pelletier |
| sedimentary_thickness_max | Maximum upland, valley bottom and lowland sedimentary deposit thickness | m | Pelletier |
| sedimentary_thickness_std | Standard deviation of upland, valley bottom and lowland sedimentary deposit thickness | m | Pelletier |
| average_thickness_min | Minimum average soil and sedimentary deposit thicknesses | m | Pelletier |
| average_thickness_mean | Mean average soil and sedimentary deposit thicknesses | m | Pelletier |
| average_thickness_max | Maximum average soil and sedimentary deposit thicknesses | m | Pelletier |
| average_thickness_std | Standard deviation of average soil and sedimentary deposit thicknesses | m | Pelletier |
| porosity_min | Minimum porosity | − | GLHYMPS |
| porosity_mean | Mean porosity | − | GLHYMPS |
| porosity_max | Maximum porosity | − | GLHYMPS |
| porosity_std | Standard deviation of porosity | − | GLHYMPS |
| log_permeability_min | Minimum permeability[2] | $m^2$ | GLHYMPS |
| log_permeability_mean | Mean permeability | $m^2$ | GLHYMPS |
| log_permeability_max | Maximum permeability | $m^2$ | GLHYMPS |
| log_permeability_std | Standard deviation of permeability | $m^2$ | GLHYMPS |

[1] For definitions and user notes, see: https://daac.ornl.gov/SOILS/guides/Global_Soil_Regolith_Sediment.html (last access: 2024-03-06).

[2] Note that permeability $k$ in the GLHYMPS database is given as $log10(k)$, due to the many decimals places otherwise needed.



**Table A9.** Subsurface attributes - continued: properties derived from Soilgrids data. Attributes are provided at six depths {D}: $0-5\text{cm}$, $5-15\text{cm}$, $15-30\text{cm}$, $30-60\text{cm}$, $60-100\text{cm}$ and $100-200\text{cm}$, and for the Soilgrids *mean* (abbreviated in the table as *{M}*) and *uncertainty* (*{U}* in the table) data fields. The mean values may be seen as expected values for a given grid cell, while the uncertainty is defined as the 90% prediction interval divided by the median value for the cell[1].

| Attribute | Description | Units | Data source |
|---|---|---|---|
| bdod_{M/U}_{D}_min | Minimum bulk density of fine earth | $\text{cg cm}^{-3}$ | Soilgrids |
| bdod_{M/U}_{D}_mean | Mean bulk density of fine earth | $\text{cg cm}^{-3}$ | Soilgrids |
| bdod_{M/U}_{D}_max | Maximum bulk density of fine earth | $\text{cg cm}^{-3}$ | Soilgrids |
| bdod_{M/U}_{D}_std | Standard deviation of bulk density of fine earth | $\text{cg cm}^{-3}$ | Soilgrids |
| cfvo_{M/U}_{D}_min | Minimum volumetric content of fragments > 2 mm | $\text{cm}^3\,\text{dm}^{-3}$ | Soilgrids |
| cfvo_{M/U}_{D}_mean | Mean volumetric content of fragments > 2 mm | $\text{cm}^3\,\text{dm}^{-3}$ | Soilgrids |
| cfvo_{M/U}_{D}_max | Maximum volumetric content of fragments > 2 mm | $\text{cm}^3\,\text{dm}^{-3}$ | Soilgrids |
| cfvo_{M/U}_{D}_std | Standard deviation of volumetric content of fragments > 2 mm | $\text{cm}^3\,\text{dm}^{-3}$ | Soilgrids |
| clay_{M/U}_{D}_min | Minimum clay fraction | $\text{g kg}^{-1}$ | Soilgrids |
| clay_{M/U}_{D}_mean | Mean clay fraction | $\text{g kg}^{-1}$ | Soilgrids |
| clay_{M/U}_{D}_max | Maximum clay fraction | $\text{g kg}^{-1}$ | Soilgrids |
| clay_{M/U}_{D}_std | Standard deviation of clay fraction | $\text{g kg}^{-1}$ | Soilgrids |
| sand_{M/U}_{D}_min | Minimum sand fraction | $\text{g kg}^{-1}$ | Soilgrids |
| sand_{M/U}_{D}_mean | Mean sand fraction | $\text{g kg}^{-1}$ | Soilgrids |
| sand_{M/U}_{D}_max | Maximum sand fraction | $\text{g kg}^{-1}$ | Soilgrids |
| sand_{M/U}_{D}_std | Standard deviation of sand fraction | $\text{g kg}^{-1}$ | Soilgrids |
| silt_{M/U}_{D}_min | Minimum silt fraction | $\text{g kg}^{-1}$ | Soilgrids |
| silt_{M/U}_{D}_mean | Mean silt fraction | $\text{g kg}^{-1}$ | Soilgrids |
| silt_{M/U}_{D}_max | Maximum silt fraction | $\text{g kg}^{-1}$ | Soilgrids |
| silt_{M/U}_{D}_std | Standard deviation of silt fraction | $\text{g kg}^{-1}$ | Soilgrids |
| soc_{M/U}_{D}_min | Minimum organic carbon content | $\text{dg kg}^{-1}$ | Soilgrids |
| soc_{M/U}_{D}_mean | Mean organic carbon content | $\text{dg kg}^{-1}$ | Soilgrids |
| soc_{M/U}_{D}_max | Maximum organic carbon content | $\text{dg kg}^{-1}$ | Soilgrids |
| soc_{M/U}_{D}_std | Standard deviation of organic carbon content | $\text{dg kg}^{-1}$ | Soilgrids |
| porosity_{M}_{D}_min | Minimum soil porosity | $-$ | Soilgrids |
| porosity_{M}_{D}_mean | Mean soil porosity | $-$ | Soilgrids |
| porosity_{M}_{D}_max | Maximum soil porosity | $-$ | Soilgrids |
| porosity_{M}_{D}_std | Standard deviation of soil porosity | $-$ | Soilgrids |
| conductivity_{M}_{D}_min | Minimum soil conductivity | $\text{cm hr}^{-1}$ | Soilgrids |
| conductivity_{M}_{D}_mean | Harmonic mean of soil conductivity[2] | $\text{cm hr}^{-1}$ | Soilgrids |
| conductivity_{M}_{D}_max | Maximum soil conductivity | $\text{cm hr}^{-1}$ | Soilgrids |
| conductivity_{M}_{D}_std | Standard deviation of soil conductivity[2] | $\text{cm hr}^{-1}$ | Soilgrids |

[1] See: https://www.isric.org/explore/soilgrids/faq-soilgrids (last access: 2024-03-07).



**Table A10.** Hydrologic signatures. Note that streamflow observations have been converted from $\mathrm{m^3\,s^{-1}}$ to $\mathrm{mm\,day^{-1}}$ using the basin areas of our newly delineated basin outlines. Please note the uncertainty in these area estimates (Figure 2). For each signature, we calculated a sequence of yearly values, and then found the mean and standard deviation across all years for which data was available.

| Attribute | Description | Units | Data source |
|---|---|---|---|
| num_years_hyd | Years of daily data used to calculate signatures | years | - |
| daily_discharge_mean | Mean daily discharge | $\mathrm{mm\,day^{-1}}$ | USGS/WSC |
| daily_discharge_std | Standard deviation of daily discharge | $\mathrm{mm\,day^{-1}}$ | USGS/WSC |
| daily_discharge_mean_month_{X} | Mean daily discharge for month X | $\mathrm{mm\,day^{-1}}$ | USGS/WSC |
| daily_discharge_std_month_{X} | Standard deviation of average daily discharge in month X | $\mathrm{mm\,day^{-1}}$ | USGS/WSC |
| runoff_ratio_mean | Ratio of mean daily discharge to mean daily precipitation | $-$ | USGS/WSC, RDRS |
| runoff_ratio_std | Ratio of mean daily discharge to mean daily precipitation | $-$ | USGS/WSC, RDRS |
| streamflow_elasticity | Streamflow sensitivity to changes in precipitation[1] | $-$ | USGS/WSC, RDRS |
| slope_fdc_mean | Slope of the log-transformed flow duration curve ($33^{th}$ to $66^{th}$ percentile)[5] | $-$ | USGS/WSC |
| slope_fdc_std | Standard deviation of the log-transformed flow duration curve[5] | $-$ | USGS/WSC |
| bfi_mean | Mean baseflow index (ratio of mean daily baseflow[2] to mean daily discharge) | $-$ | USGS/WSC |
| bfi_std | Standard deviation of baseflow index | $-$ | USGS/WSC |
| hfd_mean | Circular mean half flow date[3] | day of year | USGS/WSC |
| hfd_std | Circular standard deviation of half flow dates | days | USGS/WSC |
| q{Y}_mean[4] | Mean Y% flow quantile, where q1 are low flows | $\mathrm{mm\,day^{-1}}$ | USGS/WSC |
| q{Y}_std[4] | Standard deviation of Y% flow quantiles | $\mathrm{mm\,day^{-1}}$ | USGS/WSC |

[1] Calculated as described in Eq. 7 of Sankarasubramanian et al. (2001), with the modification described in Table 3 in Addor et al. (2017a).

[2] Calculated from time series of baseflow derived using the Eckhardt (2005) digital filter method, as recommend and implemented by Xie et al. (2020).

[3] Calculated as the day when cumulative flow in a water year passes half the total flow for that water year.

[4] Y is one of: [0.01, 0.05, 0.10, 0.25, 0.50, 0.75, 0.90. 0.95, 0.99].

[5] In cases with zero flows, 0.1% of the mean flow is added to prevent issues with calculating the logarithm. Time steps with missing flow observations are removed from the calculation.





**Table A11.** Hydrologic signatures - continued: frequency, duration and timing of high and low flow events.

| Attribute | Description | Units | Data source |
|---|---|---|---|
| no_flow_freq | Frequency of no flow days | days year$^{-1}$ | USGS/WSC |
| no_flow_dur_mean | Mean duration of no flow days | days | USGS/WSC |
| no_flow_dur_median | Median duration of no flow days | days | USGS/WSC |
| no_flow_dur_skew | Skew of no flow day durations | – | USGS/WSC |
| no_flow_dur_kurtosis | Kurtosis of no flow day durations | – | USGS/WSC |
| no_flow_timing | Season during which most no flow days occur | season | USGS/WSC |
| low_flow_freq | Frequency of low flow days (< 0.2 times the mean daily flow)[1] | days year$^{-1}$ | USGS/WSC |
| low_flow_dur_mean | Mean duration of low flow days (< 0.2 times the mean daily flow) | days | USGS/WSC |
| low_flow_dur_median | Median duration of low flow days (< 0.2 times the mean daily flow) | days | USGS/WSC |
| low_flow_dur_skew | Skew of low flow day durations (< 0.2 times the mean daily flow) | – | USGS/WSC |
| low_flow_dur_kurtosis | Kurtosis of low flow day durations (< 0.2 times the mean daily flow) | – | USGS/WSC |
| low_flow_timing | Season during which most low flow days occur (< 0.2 times the mean daily flow) | season | USGS/WSC |
| high_flow_freq | Frequency of high flow days (> 9 times the median daily flow)[1] | days year$^{-1}$ | USGS/WSC |
| high_flow_dur_mean | Mean duration of high flow days (> 9 times the median daily flow) | days | USGS/WSC |
| high_flow_dur_median | Median duration of high flow days (> 9 times the median daily flow) | days | USGS/WSC |
| high_flow_dur_skew | Skew of high flow day durations (> 9 times the median daily flow) | – | USGS/WSC |
| high_flow_dur_kurtosis | Kurtosis of high flow day durations (> 9 times the median daily flow) | – | USGS/WSC |
| high_flow_timing | Season during which most high flow days occur (> 9 times the median daily flow) | season | USGS/WSC |

[1] For consistency, we use the same definitions of dry and wet days as used in Addor et al. (2017a).



*Author contributions.* MC developed the idea for this data set and secured funding; AP provided general guidance during the project and early feedback on paper drafts; NC provided guidance on geospatial data products; LRT provided assistance with geospatial data processing coding; CT tested an early version of the data set and discovered various processing errors; KK provided assistance with forcing data

subsetting; WK developed the methodology, created the code, performed the data processing and wrote the initial draft of this paper; the paper was finalized with contributions of all co-authors.

*Competing interests.* The authors declare they have no competing interests.

*Disclaimer.* The data set described in this paper is provided in the hopes that it will be useful, but without any guarantee of correctness or fitness-for-purpose. See the license terms for full details.

*Acknowledgements.* We express our thanks to the United States Geological Survey and the Water Survey of Canada for their assistance in understanding how both agencies deal with time zones and timestamps in their data. We are grateful to Chris Marsh for pointing out some nuances about wind direction definitions, to Guoqiang Tang for providing details about the way timestamps in the EM-Earth data must be interpreted, and to Frederik Kratzert for pointing out an issue with duplicated basin IDs. We also happily acknowledge the help of Louise Arnal, Chris Marsh, and Gaby Gründemann for specific suggestions about our figures. We gratefully acknowledge the continued support

with computational resources from the Global Institute for Water Security. This project received funding under award NA22NWS4320003 from the NOAA Cooperative Institute Program. The statements, findings, conclusions, and recommendations are those of the author(s) and do not necessarily reflect the views of NOAA.



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
