# Peer review of "Catchment Attributes and MEteorology for Large-Sample SPATially distributed analysis (CAMELS-SPAT): Streamflow observations, forcing data and geospatial data for hydrologic studies across North America"

_EGUsphere, 2025_

## Referee Comment (RC2)

Comment for "Catchment Attributes and Meteorology for Large-Sample SPATially distributed analysis (CAMELS-SPAT): Streamflow observations, forcing data and geospatial data for hydrologic studies across North America" by Knoben et al.

General comments:

This study meaningfully contributes to an improvement and expansion of an important testbed for hydrologic modeling community, CAMELS, by enabling a spatial-distributed capacity. This is a significant undertaking which involves huge amount of work, including data selection, pre-processing, quality control, optimization, and validation. I can see the great potential and popularities of the resulting CAMELS-SPAT datasets being widely used in the hydrological modeling community as a testbed. I especially appreciate the following two points about this manuscript

- I appreciate authors combining methods and outcomes, which enhance the readability given the complexity of this manuscript.
- Kudos to authors about their effort in getting the details right, especially the differences in backward-looking and forward-looking timestamps for meteorological forcing datasets, which are usually overlooked.

However, I have noticed several caveats or limitations that I would like the clarifications from the authors.

- First, there is no single basins in Alaska, which is an important area for studying hydrologic change. I completely understand the data sparsity in Alaska, but I am still slightly surprising that there is no single catchment in Alaska that fit into the criteria for selection. Can the authors explain what is the biggest problems with the data in Alaska that they did not make it to the list?
- Since the authors use soil properties from SOILGRIDs, which means that the soil properties only available to the top 2 meters. For hydrologic modeling applications with deep soil columns, can the authors provide suggestions concerning the data source or assumptions of deeper-than-2-meter soil properties?
- I got slightly confused when I started reading Section 3, because I thought the catchment attributes, such as climatology, vegetation, land cover, etc., have already been discussed in Section 2.5. Why do we need another section to discuss it? After reading it through, apparently Section 3 provides a more in-depth analysis of the catchment attributes. However, I would still recommend the authors making some changes to help the readers understand the flow of this manuscript. A possible way to do so is to add a short description of paper structure at the end of Introduction.

Specific comments:
L45: Please cite the original VIC paper
- Liang, X., Lettenmaier, D. P., Wood, E. F., & Burges, S. J. (1994). A simple hydrologically based model of land surface water and energy fluxes for general circulation models. Journal of Geophysical Research: Atmospheres, 99(D7), 14415-14428.

L62: It is unclear to me how to differentiate 1) sub-basin, 2) basin, and 3) catchment in this context. Is basin equivalent to catchment in this sentence? Do sub-basins mean delineation of entire basin or catchment to different vector-based hydrologic response units (HUCs)?

L70: "Native" was used to describe the data characteristics multiple times in this paragraph. I guess "native" would refer to that the authors keep the "original" spatial resolution of the source met forcing data or other geospatial data sets. I would recommend the authors add one sentence defining "native", deleting the parenthesis in lines 62, 70, 72.

I am not sure about it but the subsections 2.X.1 with title "context" seems a bit redundant. If you do not provide this subtitle, readers could still know this part is the context or background information for this section. The subsections "2.X.2" with title "Methods and outcomes" seems ok.

L102: Delete "for" in this sentence, i.e., " e.g. agricultural and industrial use, …"

L223: Kudos for accounting for daylight savings.

L229-233: Kudos for explaining what the timestamp represents.

L277: "Days" should be plural in the context, i.e., "between the preceding and following day**s**"

L287: What is the area of interest in this study? Do authors refer to all CAMELS-SPAT basins?

L317: What do "whole hours" mean? Please clarify

L389: Do you mean $5^{th}$ percentile, rather than $0.05^{th}$ percentile? Maybe a typo?

The title for Section 3 is confusing. I thought Section 2.5 already covers the catchment attributes.

L407-408: "Only" and "rarely" are redundant in this sentence, i.e., "…the existing literature **only rarely** uses catchment descriptors…" Please delete one of them.

L449. Please spell out "m.a.s.l.".

L454:  Please use SI and avoid using hectare (ha).

L497: Does the author mean "skewness" values?

L587: Did the authors mean the selection of attributes **for data-driven models** in this context? In most process-based hydrologic models, the attributes that serve as input are usually pre-defined and not much selection should be required.

---

## Author Response (AR1)

Dear reviewers, dear editor,

Thank you for your work on assessing our paper. It is rather long, and we appreciate your efforts. Please find our responses to your comments below, in blue.

We appreciate the editor's invitation to make further changes that we deemed helpful and have taken the opportunity to do so. In addition to the clarifications requested by the reviewers, we have made the following changes to the manuscript and associated data resource (note that the data set DOI has changed accordingly: 10.20383/103.01306).

**Enhancements:**

We substantially expanded the time period covered by hourly data for the Canadian gauges, which was possible because the Water Survey of Canada has recently made a longer part of their historical records openly available. For most Canadian basins, this means the time period for which hourly values are available has increased from 18 months to over a decade. This had the added benefit of replacing a large number of hourly streamflow observations with the "provisional" tag with "approved" values, and this particularly affects discharge values under ice-covered conditions (see figure below for an example; "File 1" refers to the old file, "File 2" to the new file; bottom plot shows how the "provisional" data values from file 1 compare to the "approved" values in file 2 under regular [blue] and ice [yellow] conditions). Metadata, figures and text in the paper were updated accordingly.

- We added summary sheets of the data for each catchment to the data repository. These are high-level data summaries that can help users get a quick idea of the sort of basin they are looking at. We added a brief mention and explanation of these to the main manuscript (new section 4.2.1) and added an example sheet to the SI.
- We moved the "known issues" record from the data repository onto the code repository. The benefit of this is that it is much easier to keep this record updated in case new issues are identified. The "known\_issues.txt" file on the data repository now points the user to the GitHub version instead: <a href="https://github.com/CH-Earth/camels\_spat/issues/42">https://github.com/CH-Earth/camels\_spat/issues/42</a>
- We added two shapefiles that contain (1) all station locations, and (2) all individual lumped basin outlines. We envision these will be useful for visualization purposes.
- We updated the data README to include brief descriptions of various extra variables we were able to provide as part of the river shapefiles.

**Corrections:**

- We corrected an inconsistency with timestamps in all observed streamflow files.
   Previously, the time stamps and time bounds were specified in UTC, and this has been updated to be LST to match the description in the paper.
  - For daily flow values, this has no consequences beyond a change in time step and time bounds values.
  - o For hourly flow values, the older files also used a different (simpler) averaging procedure to derive hourly values from sub-hourly observations than the new files. This means hourly flow values are thus somewhat different in these new files. Differences are minor however (typically well below 1% for mean flow and standard deviation of flows, as well as Pearson correlation coefficients still being > 0.99 in almost all cases). All streamflow statistics reported in the paper and attributes were calculated from the daily files and are thus unaffected.
- We identified a minor issue affecting the SOILGRIDS maps. In 485 basins the maps for specific variables were aligned slightly differently compared to other maps for the same basin. To ease processing for users we ensured all maps are aligned. Values in the maps remain unchanged.
- We identified 30 basins where part of the RDRS forcing data time series was missing. We corrected this and updated the relevant attribute files accordingly, as well as Figure 5 in the paper that shows various climate attributes.
- We corrected the units of the rainfall-runoff attributes and certain forcing attributes in all attribute files. The units of the rainfall-runoff attributes somehow appeared as mm d-1 month-1 instead of dimensionless. For certain forcing attributes the time

- period as missing (e.g., mean annual statistics would be listed with units [mm] instead of [mm year-1]). Values for the attributes remain unchanged.
- We identified and corrected an issue in the connectivity of our river polygons, specifically in polygons just upstream and downstream of river segments that we had to split to account for nested gauges. These river segments now point to the correct split polygon part immediately upstream/downstream from themselves.
  - We added values for "upstream area" and "slope" to river segments we had to split, to bring these in line with the information already available for nonsplit segments.
- We identified and resolved an issue in the calculation of missing flow data (Figure 3, Figure A1) that results from a misunderstanding of how the data providers deal with missing values in their data products. Both figures and the relevant text have been updated.
- We identified and corrected a mistake in the units of Daymet data in Table 3.
- We cleaned up a variety of typo's in the main paper.

Kind regards,

Wouter Knoben

**Reviewer 1**

Overall, the authors did an excellent job explaining the processing of the CAMEL-SPAT geospatial dataset. The narrative is clear and provides sufficient details and explanation to understand the basic process for preparing the dataset(s). The tables are very detailed and helpful in visualizing the different models, model requirements, and the dataset(s).

Thank you for these kind words. It is good to see you see merit in this work. Please find replies to your individual comments below. Briefly, we see the value in your requested clarifications and will implement them as suggested in all cases apart from the request for a CDF of mean flows. This information is already available in the "streamflow signature" section of the paper and we thus prefer to refer the reader there instead of adding another figure to an already long manuscript.

Overall, I have relatively minor edits, as explained below:

Abstract

Line 14: small typo (This data set)

Changed.

Line 1: I think the first line of the abstract should make it clear that the dataset builds on the CAMELS dataset; perhaps expand on "new dataset" such as "new dataset that builds on the CAMELs dataset".

Added. New text in bold:

We **build on the existing CAMELS data set to** present a new data set aimed at hydrologic studies across North America, with a particular focus on facilitating spatially distributed studies.

Line 171: small type "area"

Changed.

Line 185: Please provide more details on the accuracy assessment for the basin methods. Specifically, provide details on what you mean by "evidence suggests". What statistical/quantitative measurement did you use to identify your "confidence ratings"?

This refers to the approach described in the preceding paragraph:

We identified those cases through a combination of accuracy metrics (area comparison between new basin delineation and reported reference area(s), and percentage overlap between new basin delineation and reference polygon if any were available), and visual inspection of the new basin delineation, reference polygon, underlying MERIT Hydro data grids, and satellite images.

We have added a brief clarification in the later lines (changes in bold):

Figure 2 shows the resulting polygons for the 1426 basins that form the final CAMELS-SPAT data set, with colors indicating the confidence ratings we assigned based on the checks listed previously (i.e., automated overlap and area checks, as well as manual inspection of polygons and satellite images).

Figure 3: Please clarify a few elements of this figure in the caption: The figure has 3 colors (pink, blue, maroon) – please explain maroon. Also, explain the record length and how it differs from missing values in the caption and somewhere in the paragraph from line 240-249.

The maroon appears where the pink and blue bars overlap. We have added the following sentence to the figure caption to clarify this:

Please note that the colors are partly transparent, and that overlaps between the *record length* and *missing values* bars will appear as dark red.

With respect to record length and missing values, caption 3 already contains the following text:

"Flow data availability for gauges included in CAMELS-SPAT. Record length refers to the period between the first publicly available flow record for a given station, and its last. Missing values occur within this record period and are given here in the same units as the record length itself."

This seems exactly the explanation the reviewer is asking for in the second part of their comment, and it is not fully clear to us what needs to be clarified here. We added an explanation to the text where Figure 3 is first mentioned (new text in bold):

Figure 3 shows aggregated flow data availability for the 1426 catchments included in the CAMELS-SPAT data set, with total record length in blue (number of years between first and last available streamflow observation) and missing values in red (number of years within the record length for which no observations are available).

Line 210 – Methods and Outcomes section for Streamflow observations: I would recommend creating a CDF figure similar to Figure 2 in Newman et al., 2015 (see below) in order to show the streamflow distribution of the new dataset that you present here.

Data citation: A. Newman; K. Sampson; M. P. Clark; A. Bock; R. J. Viger; D. Blodgett, 2014. A large-sample watershed-scale hydrometeorological dataset for the contiguous USA. Boulder, CO: UCAR/NCAR. https://dx.doi.org/10.5065/D6MW2F4D

This CDF in Newman et al. (2015) shows the distribution of mean annual runoff. In our case, this information can be found in Figure 9a (though presented as a histogram and on a map). We have added a forward-looking reference to the end of this section, so the reader knows this information is available in the section on streamflow signatures:

Some further statistics about the streamflow regimes available in CAMELS-SPAT are discussed in Section 3.5.

348: I'd recommend also referencing Figure 1, as that visualizes the "processing steps" referred to in this line.

Added.

Figure 7 caption: provide more details similar to those presented in the Figure 5 and 6 captions. For example, I recommend at the least, including the dataset used to create the figures and how/where the statistics were derived.

Agreed. We add the bolded text below:

Figure 7. Selection of vegetation attributes. (a, b) The mean and standard deviation of forest height within each basin are derived from the Global Land Cover and Land Use Change data set and are shown here for the year 2020. (c, d) Leaf Area Index values are derived from the MODIS MCD15A2H.061 data set and are shown here as long-term averages values for February and August.

Figure 8 caption: Same as above (e.g. dataset and statistics explanation), but also include explanation of prediction uncertainty

Agreed. Proposed new text in bold:

Figure 8. Selection of subsurface attributes. (a-d) Properties derived from the SOILGRIDS 2.0 data set through spatial averaging for each catchment. (a, b) Mean and spatial standard deviation of sand content in the top SOILGRIDS layer. (c, d) Mean and spatial standard deviation of sand content uncertainty, defined as the ratio between the 90-percentile prediction interval and the median prediction ( $Q_{95}-Q_{05}$ )/ $Q_{50}$ . (e, f) Mean and spatial standard deviation of sedimentary deposit thickness estimates in the Pelletier data set.

For completeness, we also slightly expanded the caption of Figure 9:

Figure 9. Selection of hydrologic signatures, derived from timeseries of daily data provided by USGS and WSC.

**Reviewer 2**

**General comments:**

This study meaningfully contributes to an improvement and expansion of an important testbed for hydrologic modeling community, CAMELS, by enabling a spatial-distributed capacity. This is a significant undertaking which involves huge amount of work, including data selection, pre-processing, quality control, optimization, and validation. I can see the great potential and popularities of the resulting CAMELS-SPAT datasets being widely used in the hydrological modeling community as a testbed. I especially appreciate the following two points about this manuscript:

- I appreciate authors combining methods and outcomes, which enhance the readability given the complexity of this manuscript.
- Kudos to authors about their effort in getting the details right, especially the differences in backward-looking and forward-looking timestamps for meteorological forcing datasets, which are usually overlooked.

Thank you for these kind words. It's nice to see this appreciation of our efforts.

However, I have noticed several caveats or limitations that I would like the clarifications from the authors.

• First, there is no single basins in Alaska, which is an important area for studying hydrologic change. I completely understand the data sparsity in Alaska, but I am still slightly surprising that there is no single catchment in Alaska that fit into the criteria for selection. Can the authors explain what is the biggest problems with the data in Alaska that they did not make it to the list?

This is an artefact of our methodology: by building the US part of our data set on the existing CAMELS data set, we are limited to basins in the contiguous US (there are also no basins for the Hawaiian Islands, Puerto Rico and the US Virgin Islands for example). This choice for CONUS-only basins is not explicitly motivated in either Newman et al. (2015) or Addor et al. (2019), but (presumably) stems from the fact that Newman et al. (2015) selected daymet as their primary forcing data product:

"Daymet is a daily, gridded (1 × 1 km) data set over the CONUS and southern Canada and is available from 1980 to present."

The timing of references etc. suggests that this refers to Daymet v2 (<a href="https://daac.ornl.gov/DAYMET/guides/Daymet\_mosaics.html">https://daac.ornl.gov/DAYMET/guides/Daymet\_mosaics.html</a>), which at the time was not available for Alaska and the islands regions.

There are no data quality reasons that we are aware of that would prevent inclusion of basins from Alaska (several are part of the HCDN-2009 streamflow network that Newman et al., 2015, used for their basin selection; see Lins, 2012).

In terms of hydrologic variability, we did include various basins in northern Canada (particularly in the Yukon and Northwest Territories, with a handful of extra basins in the Nunavut Territories). These should at least be somewhat similar to basins found in Alaska in terms of hydrologic processes and sensitivity to hydrologic change.

The spatial coverage of our primary forcing product (RDRS v2.1) still prevents inclusion of the Hawaiian Islands, Puerto Rico, and US Virgin Islands.

We added a brief explanation of this discussion to the section on basin selection (new text in bold):

For basins in the United States, we rely on the basin selection made by Newman et al. (2015) that was used for the CAMELS data set (Addor et al., 2017a). This ensures that some level of comparison between outcomes of studies using either CAMELS or CAMELS-SPAT is possible. We refer the reader to Section 2.1 in Newman et al. (2015) for a description of the criteria used to create this selection of 671 basins, and note that, despite meeting these criteria, no basins in Alaska, Hawaii and Puerto Rico were included in the original CAMELS data set due to limited spatial coverage of the Daymet data at the time. Our primary forcing data set (see Section 2.4) does not have the coverage to include basins in Hawaii or Puerto Rico, but cold region processes as may be found in Alaska are covered by our selection of Canadian basins.

Lins, H. F.: USGS Hydro-Climatic Data Network 2009 (HCDN-2009), US Geological Survey, Fact Sheet 2012-3047, Reston VA, USA, 2012

Since the authors use soil properties from SOILGRIDs, which means that the soil
properties only available to the top 2 meters. For hydrologic modeling applications
with deep soil columns, can the authors provide suggestions concerning the data
source or assumptions of deeper-than-2-meter soil properties?

This is difficult to do in a general sense, because what may be appropriate in one model may not be appropriate (or even possible) in the next. We have added a short new section to our discussion section "Guidelines for practical use" that at least outlines this problem, though we would like to avoid appearing to endorse one method over another.

**New text:**

**4.2.4 Combing soil depth and soil properties estimates**

CAMELS-SPAT contains both estimates of soil depth (derived from the Pelletier data set; Pelletier et al. (2016a, b)) and soil properties (derived from the SOILGRIDS 2.0 data set; Poggio et al. (2021)). Because the SOILGRIDS data assumes a uniform depth of 2.0 meters everywhere, soil properties will thus be unknown for actual soil depths greater than 2 meter, or incorrectly provided for actual soil depths less than 2 meters. For estimated depths below 2 meters, an appropriate approach may be to only use the SOILGRIDS layers that correspond to the estimated soil depth. For estimated soil depths greater than 2 meters, recommendations are more difficult to provide. Appropriate approaches may be the derivation of pedotransfer

functions, or reliance on simple assumptions that extend the available layer information to deeper depths.

• I got slightly confused when I started reading Section 3, because I thought the catchment attributes, such as climatology, vegetation, land cover, etc., have already been discussed in Section 2.5. Why do we need another section to discuss it? After reading it through, apparently Section 3 provides a more in-depth analysis of the catchment attributes. However, I would still recommend the authors making some changes to help the readers understand the flow of this manuscript. A possible way to do so is to add a short description of paper structure at the end of Introduction.

We have added this paper structure description as requested and tried to clarify up front what the difference between Section 2.5 (downloading and processing of geospatial data) and Section 3 (deriving attributes from the now-processed geospatial data) is. New text:

This paper is structured as follows. Section 2 starts by outlining our design considerations for this data set, followed by five longer sub-sections that describe the methods and outcomes of our basin selection (Section 2.1), basin delineation (Section 2.2), streamflow observation processing (Section 2.3), forcing data processing (Section 2.4), and geospatial data processing procedures (Section 2.5). Section 3 then provides details on how we used the geospatial data to derive over 1100 statistical descriptors of each basin, also known as catchment attributes. Section 4 has various recommendations for data providers based on our experiences building the CAMELS-SPAT data set (Section 4.1), as well as various recommendations for data users based on our expectations of how the CAMELS-SPAT data might be used (Section 4.2). Section 4 also contains some thoughts about extension of the data set to new regions (Section 4.3), and notes on the data set structure and size (Section 4.4). A summary and conclusions are given in Section 5.

Specific comments:

L45: Please cite the original VIC paper

 Liang, X., Lettenmaier, D. P., Wood, E. F., & Burges, S. J. (1994). A simple hydrologically based model of land surface water and energy fluxes for general circulation models. Journal of Geophysical Research: Atmospheres, 99(D7), 14415-14428.

Done. We have kept the Hamman et al. (2018) reference in Table 2 though, because we used it to find the information listed in the Table.

L62: It is unclear to me how to differentiate 1) sub-basin, 2) basin, and 3) catchment in this context. Is basin equivalent to catchment in this sentence? Do sub-basins mean delineation of entire basin or catchment to different vector-based hydrologic response units (HUCs)?

Yes to both. We triedy to clarify this as follows:

First, we provide the CAMELS-SPAT data at three spatial resolutions: (1) at its original gridded resolution; (2) spatially averaged at the sub-basin level (smaller areas that subdivide the area upstream of each gauge to facilitate semi-distributed modeling); (3) spatially averaged at the basin level (equivalent to how catchments are treated as lumped entities in the original CAMELS data set).

L70: "Native" was used to describe the data characteristics multiple times in this paragraph. I guess "native" would refer to that the authors keep the "original" spatial resolution of the source met forcing data or other geospatial data sets. I would recommend the authors add one sentence defining "native", deleting the parenthesis in lines 62, 70, 72.

It's probably clearer to just call it "original [spatial resolution]" like done in this comment, so we have changed all instances of "native" to "original" in the text.

I am not sure about it but the subsections 2.X.1 with title "context" seems a bit redundant. If you do not provide this subtitle, readers could still know this part is the context or background information for this section. The subsections "2.X.2" with title "Methods and outcomes" seems ok.

Some division between the more general "context" part and the actual work and results seems indeed helpful. Having just a single sub-heading ("2.x.1 Methods and outcomes") within each larger 2.x section looks a little odd to us, and we have therefore elected to keep the "2.x.1. Context" headers intact.

L102: Delete "for" in this sentence, i.e., "for e.g. agricultural and industrial use, ..."

**Changed:**

"... surface and groundwater abstractions (e.g., agricultural and industrial use), for which abstraction and return volumes are typically unknown."

L223: Kudos for accounting for daylight savings.

L229-233: Kudos for explaining what the timestamp represents.

L277: "Days" should be plural in the context, i.e., "between the preceding and following days"

Changed.

L287: What is the area of interest in this study? Do authors refer to all CAMELS-SPAT basins?

In this case, this refers to the North America continent. We'll clarify this as follows:

Previous work has shown that using station-based precipitation and temperature data from EM-Earth provides better modeling results for **the North American continent** than using ERA5 alone (Rakovec et al., 2023).

L317: What do "whole hours" mean? Please clarify

What this means is that the data is only available at the top of each hour. We've clarified this as follows:

The time series of all forcing data products only provide values at the top of each hour (12:00, 1:00, etc.), and thus cannot easily be converted to NST.

L389: Do you mean 5th percentile, rather than 0.05th percentile? Maybe a typo?

This is indeed a typo, changed to 5th.

The title for Section 3 is confusing. I thought Section 2.5 already covers the catchment attributes.

This should be clear from the extra paragraph we added to the introduction, but we've added another sentence here to remind the reader:

Existing large-sample data sets do not provide the maps of geospatial data that we include as part of CAMELS-SPAT (see Section 2.5), and instead only provide statistical summaries of such maps known as catchment attributes (for example, a data set might provide the mean catchment elevation instead of the full Digital Elevation Model).

L407-408: "Only" and "rarely" are redundant in this sentence, i.e., "...the existing literature only rarely uses catchment descriptors..." Please delete one of them.

"Only rarely" is grammatically correct English, and the "only" here emphasizes just how rarely the thing we're talking about is done. Because I think the point we're trying to make in this sentence is important, and the sentence itself is not grammatically incorrect, I would like to leave it as is.

L449. Please spell out "m.a.s.l.".

Added: "3000 meter above sea level (m.a.s.l.)"

L454: Please use SI and avoid using hectare (ha).

The HydroLAKES documentation uses hectare when discussing this threshold, and we therefore kept it here. We've replaced this with "0.1 km²" in the main text and also added this to the caption of Figure 6, but also kept the "10 ha" mention there to help readers align our text with the HydroLAKES documentation.

L497: Does the author mean "skewness" values?

Changed.

L587: Did the authors mean the selection of attributes for data-driven models in this context? In most process-based hydrologic models, the attributes that serve as input are usually pre-defined and not much selection should be required.

Yes, but also in a more general sense. Others might, for example, try to relate model performance to attribute values for regionalization, or use clustering analysis to define subgroups within our larger basin sample. We added some text to clarify this (new text in bold):

Larger numbers of attributes will, however, increase computation and analysis times for applications such as regionalization, clustering and data-driven modeling.